# Enabling reactive microscopy with MicroMator

Zachary R. Fox [1,2,3,6], Steven Fletcher[1,2,6], Achille Fraisse[1,2], Chetan Aditya [1,2], Sebastián Sosa-Carrillo[1,2], Julienne Petit[4], Sébastien Gilles[5], François Bertaux [1,2], Jakob Ruess[1,2] & Gregory Batt [1,2 ✉]

Microscopy image analysis has recently made enormous progress both in terms of accuracy and speed thanks to machine learning methods and improved computational resources. This greatly facilitates the online adaptation of microscopy experimental plans using real-time information of the observed systems and their environments. Applications in which reactiveness is needed are multifarious. Here we report MicroMator, an open and flexible software for defining and driving reactive microscopy experiments. It provides a Python software environment and an extensible set of modules that greatly facilitate the definition of events with triggers and effects interacting with the experiment. We provide a pedagogic example performing dynamic adaptation of fluorescence illumination on bacteria, and demonstrate MicroMator's potential via two challenging case studies in yeast to single-cell control and single-cell recombination, both requiring real-time tracking and light targeting at the single-cell level.

[1] Inria Paris, 2 rue Simone Iff, 75012 Paris, France. [2] Institut Pasteur, 28 rue du Docteur Roux, 75015 Paris, France. [3] Center for Nonlinear Studies, Los Alamos National Laboratory, Los Alamos, NM, USA. [4] Structural Microbiology Unit, Institut Pasteur, CNRS UMR 3528, Université de Paris, F-75015 Paris, France. [5] Inria Saclay—Ile-de-France, 1 rue H. d'Estienne d'Orves, 91120 Palaiseau, France. [6]These authors contributed equally: Zachary R Fox, Steven Fletcher. ✉email: gregory.batt@inria.fr

Software for microscopy automation is essential to support reproducible high-throughput microscopy experiments[1]. Samples can now be routinely imaged using complex spatial and temporal patterns. Yet, in the overwhelming majority of cases, executions of experiments are still cast in stone at the beginning, with little to no possibility for human or computer-driven interventions during the experiments. Applications in which reactiveness is needed are multifarious. One can for example think of detecting a loss of focus and triggering an autofocus routine, of adapting the imaging conditions to time-changing specificities of the sample, of detecting rare events and adapting the imaging routine, of detecting moving objects of interest and following them by moving the stage, or even simply of sending a message via email or messaging applications upon the detection of interesting or problematic situations that might necessitate unplanned human interventions. This situation is all the more surprising given that image analysis has recently made a giant leap in terms of accuracy and rapidity thanks to deep learning methods and software which leverage modern computational resources[2], thus opening the way for implementing elaborate protocols. Software empowering microscopy with real-time adaptation capabilities are needed to exploit the full potential of automated microscopes.

Several dedicated microscopy software solutions have been developed for applications requiring real-time analysis. This is notably the case for the efficient scanning of large and complex microscopy samples[3–7]. For applications aiming at controlling in real-time cellular processes[8–14], results are generally obtained using ad hoc software solutions. In such experiments, the goal is to perturb biomolecular processes within cells using externally controlled inputs. These inputs may be chemical[9,10,13,15] or optogenetic[8,11,12,14]. In their most elaborate form, these experiments aim to perturb or control individual cells as they grow and divide. Due the complexity that is required to coordinate the software, hardware, and biological aspects of reactive experiments, very few generic tools have been developed so far to facilitate them. One notable exception is Pycro-Manager[16]. This powerful framework is built on top of Micro-Manager, a widely-used software[17,18] controlling a large range of microscopy hardware. In Pycro-Manager, reactive protocols are built from the ground up. While this gives maximal flexibility, it also increases the difficulty to rapidly design experiments, especially for non-expert users. Moreover, no in-depth case studies demonstrating its practical applicability—and showing possible limitations—have been reported so far. One can also mention Python-Microscope, a free and open-source library that provide Python support for the high-performance control of arbitrarily complex and scalable custom microscope systems[19]. Lastly, Cheetah is a simple to use Python library to support the development of real-time cybergenetic control platforms that combines microscopy imaging and microfluidics control[20]. In its current state, the possibilities to programmatically control the microscope appear limited.

In this paper, we present MicroMator, a software solution supporting reactive microscopy experiments, provide a pedagogic example performing dynamic adaptation of fluorescence illumination on bacteria and demonstrate MicroMator's potential via two challenging case studies in yeast that require real-time tracking and light targeting at the single-cell level.

## Results

**MicroMator software.** In MicroMator, microscopy experiments are defined by a main image acquisition loop, that serves as a backbone for the experiment, and by event creation functions that implement the reactivity of the experiment. Events play a fundamental role (Fig. 1). They consist of Triggers and Effects. Examples of triggers include "at the 10th frame", "if more than 100 cells are in the field of view", and "if the fluorescence of the 3rd newborn cell exceeds 100 arb. units". Examples of effects include changing a microscope configuration, sending light in the field of view with a given pattern, actuating a microfluidic pump, starting an optimization routine, and posting "Warning: focus lost" or "Ending acquisition" messages on instant messaging

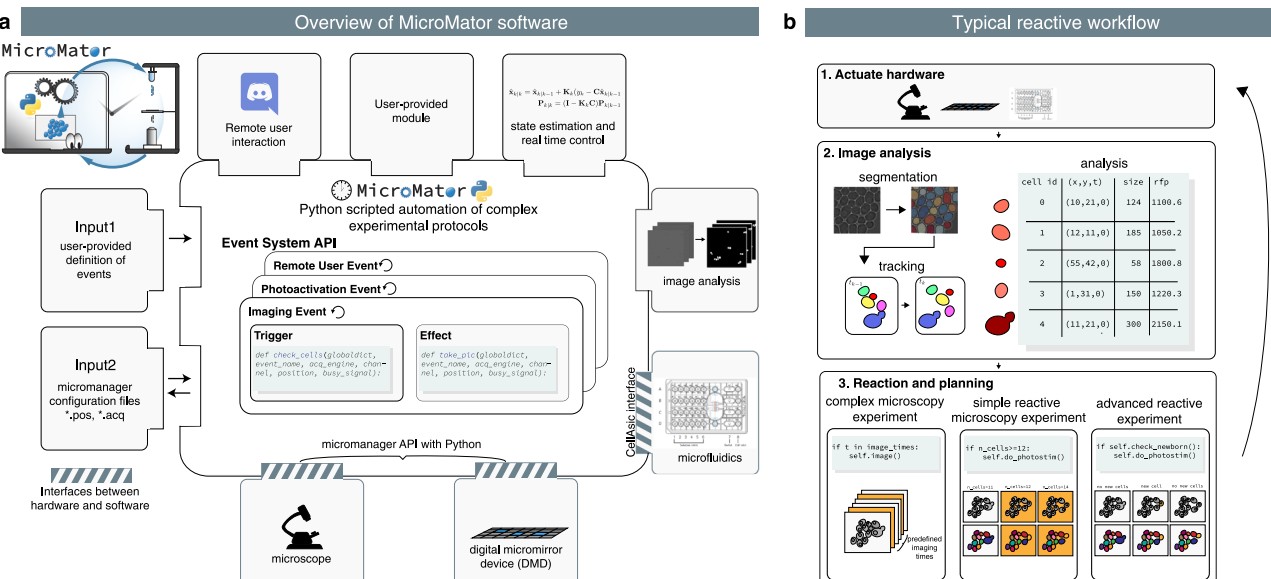

**Fig. 1 MicroMator overview. a** Modular software architecture. MicroMator consists of a core software that handles user-defined events and of an extensible set of modules that control various hardware and software aspects of microscopy experiments. It is written in the high-level programming language Python. It takes as inputs Python files defining events and Micro-Manager configuration files providing positions of interest and an imaging backbone. **b** Event-based reactive microscopy workflow. Imaging can be followed by online analysis of the samples. This typically involves segmentation, tracking, quantification of cell properties, and possibly advanced additional computations. Effects may then be triggered based on the result of the analysis. These may include the physical actuation of the hardware or the initiation of communications or of additional computations.

software platforms such as Discord. Naturally, the main acquisition loop itself can be modified by event effects in the course of the experiment.

MicroMator is written in Python 3, is open-source, and has a modular design. MicroMator is primarily a command line software that reads configuration files that define the reactive experiments. A graphical user interface (GUI) is provided as an alternative to writing command line calls. It is notably used to define which configuration files and which options should be used. Specifically, a MicroMator reactive experiment necessitates two types of configuration files. The first one provides the programmatic definition of the events. It is made of a python file that contains three functions: an event creator function, a trigger function, and an effect function. The user can use predefined functions for a few generic problems, and take inspiration from all the events that are already part of the code for specific ones. MicroMator also uses Micro-Manager-generated files that specify the positions of interest in the field of view (.pos files) and the imaging backbone of the experiment (.acq files), that is, a default image acquisition sequence with which MicroMator events will interact. They are obtained using the GUI of Micro-Manager prior to running the MicroMator experiment. Supplementary Movie 1 shows the different steps of the initialization of a MicroMator experiment. By default, events are created after each image acquisition in the main acquisition loop. Their triggers are assessed, and their effects are implemented. Because events can create events, highly complex experiments can in principle be conceived.

MicroMator has a relatively small core and relies on modules to implement most of the tasks at hand. For controlling hardware, MicroMator primarily uses the Python API of Micro-Manager pymmcore. It can also use other dedicated Python or web-based APIs provided by vendors, as done for our CellAsic ONIX microfluidic platform. In addition, various types of analysis can be performed using dedicated software modules. One can notably mention image analysis and optimization or real-time control tasks. Within MicroMator, we have developed two image analysis modules. The first one, called SegMator, is based on DeLTA[21], which uses U-Net for bright-field cell segmentation, and on TrackPy[22] for cell tracking. U-Net is a convolutional neural network with a structure that excels at image segmentation[23]. The second image analysis module we implemented is based on the generalist, deep learning-based segmentation method Cellpose[24]. Cellpose also uses the U-Net architecture. We use Cellpose to segment Corynebacteria on agar pads and SegMator/DeLTA to segment and track yeast cells in microfluidic plates. Moreover, communication modules can also be used to interface Micro-Mator with digital distribution platforms such as Discord to track experiment progress and potential issues. Lastly, MicroMator leverages Python's multiprocessing module to perform computations concurrently and possesses an extensive and customizable logging system, gathering logs of all modules in a unique file and fostering reproducible research. Naturally, the set of modules can be extended by the users to address specific needs. Novel modules can simply be interfaces with other tools or can be arbitrarily complex pieces of code. A more detailed description of the software and the different modules is available in the Supplementary Note 1. A complete example of event definition is provided in the Supplementary Note 2 for a toy problem. Using MicroMator necessitates some Python programming skills. However, we would like to stress that extending a few portions of code following a predefined architecture and having diverse examples to take inspiration from, or writing a complete reactive microscopy program from scratch are two tasks of very different complexities that involve very different programming skills. This holds for the initial development of the software and even more so for its debugging, maintenance, and reuse.

To showcase the full potential of reactive experiments performed with MicroMator, we designed experiments in which cellular processes are controlled in real-time. Single-cell stimulations are computed online based on the cell state and/or position, demonstrating that reactive loops can be implemented at the level of individual cells. These experiments are inspired by previously-published studies[11,12,14,25] and show how published studies could be repeated and further extended using generic software. However, before showing these advanced experiments, we provide a simple example that illustrates that even very simple instances of reactive experiments can help to address problems in practice. A related but even simpler example is provided in Supplementary Note 2.

**Real-time exposure adjustment in bacteria**. We consider microscopy observations of *Corynebacterium glutamicum* strains. *Corynebacterium glutamicum* is a model organism to study *Mycobacterium tuberculosis*. Cells of interest are grown on agar pads over more than 10 h. They express the cell-cycle marker Wag31 fused to the fluorescent protein mNeonGreen, and the Nile red lipolytic dye is used to stain membranes. Over several hours, the red fluorescent dye may migrate or diffuse across the agar pad, leading to a fluctuating and decreasing fluorescent signal. Moreover, the intensity of the staining varies with the different locations on the agar pad and between different agar pads. These heterogeneities and temporal variations complexify the analysis of the images: different acquisition settings are needed for different positions in the agar pad and for different time points.

To tackle this issue, we use MicroMator along with a reactive event that adapts the exposure time of the red channel acquisition at every time point to achieve a targeted fluorescence value (Fig. 2). The correction is specific to each field of view, so any number of positions can be acquired, and their respective exposure corrected during the same experiment. To measure the intracellular fluorescence, we wrote an analysis module performing bacteria segmentation in between the time points using Cellpose, a deep-learning-based generalist algorithm for cell segmentation[24]. After every image acquisition, if the mean fluorescence of the cells is too far from the chosen target, the exposure time is increased or decreased by a constant value. On Fig. 2c, we can see that MicroMator maintained the fluorescence around the chosen target through exposure adjustments. Moreover, exposure adjustments led to a less severe degradation of the ratio between the cellular fluorescence and the background fluorescence during the experiment, due to the decreasing concentration of the dye.

**Model predictive control (MPC) of gene expression at the single-cell level in yeast**. For our second application, we use the EL222 optogenetic system and the mScarletI fluorescent reporter to engineer light-responsive yeast cells (Fig. 3a). Using real-time imaging, segmentation, and cell tracking, different cells can be stimulated differently in the field of view using a digital micro-mirror device (DMD). Our goal is to implement different MPC strategies for controlling the expression levels of a protein in a cell population. We used the SegMator image analysis module to segment and track cells in the field of view in real-time. Dense fields of cells can be analyzed in a few seconds and with good accuracy (Supplementary Note 3 and Movie 2). The cellular response of our engineered cells was then characterized for different light stimulation profiles. In our experiments, only the most central part of the cell is targeted for light stimulation. This erosion of the stimulation region helps improving the precision of single-cell light stimulations in dense cell regions because of

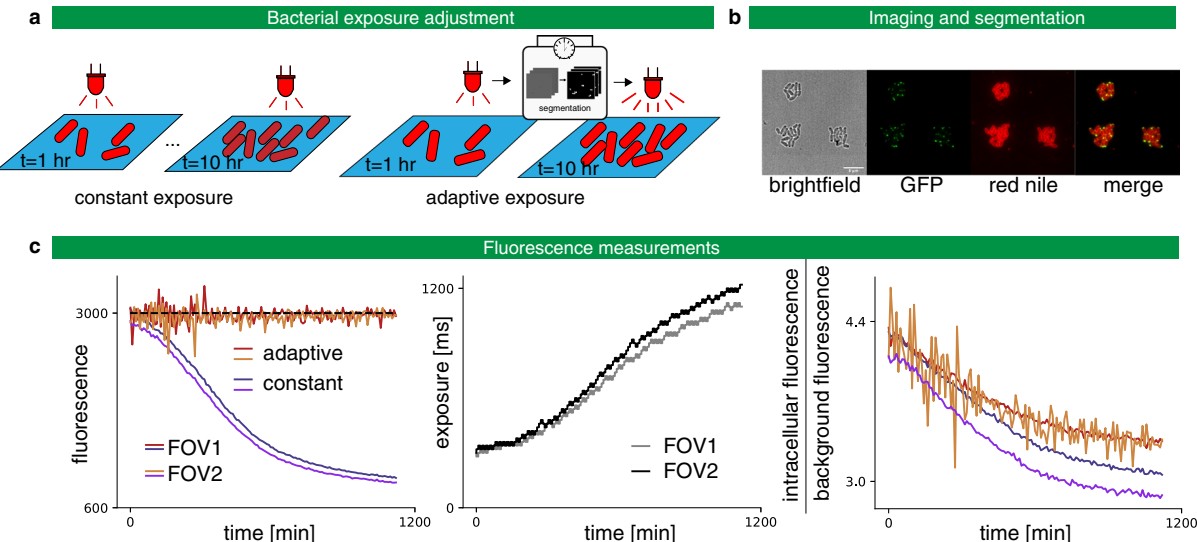

**Fig. 2 Adaptive exposure preserves imaging quality. a** Left: Classic constant exposure experiment, in which the same amount of light is sent throughout the experiment. Right: adaptive experiment, in which bacteria are segmented after each frame and the exposure is increased or decreased depending on the measured fluorescence. **b** Imaging *C. glutamicum* in brightfield and fluorescence with Wag31 fused to mNeonGreen and localized to the poles (GFP), and with red nile stain. **c** Left: In the constant exposure experiment, the fluorescence decays. For the adaptive experiment, fluorescence tracks the target value of 3000 arb. units. The experiment is performed in two fields of view (FOV) in parallel. Center: The exposure settings as a function of experiment time. Right: Signal-to-noise ratio as defined by the intracellular fluorescence divided by the background fluorescence as a function of the experiment time. A less severe degradation is obtained with the adaptative strategy.

illumination bleed-through of DMD systems (see Supplementary Note 4). Using our capacities to apply different light stimulations to the different cells in the field of view, we could produce the data set presented in Fig. 3b, and with more details in Supplementary Fig. 6, in a single experiment. We then developed and calibrated an "average cell" (deterministic) and a "single cell" (stochastic) model of light-driven gene expression (Supplementary Note 5 and Supplementary Figs. 6 and 7). We used a delay to account for the lag between the stimulation of the cell and the detection of fluorescence of the reporter. We estimated its value to be 36 min, corresponding to the estimated maturation time of mScarlet-I[26] and in accordance with the fast kinetics of the EL222 transcription factor[27]. We also note that as expected the estimated value for the protein degradation and dilution time is in good agreement with the estimated cell generation time (Supplementary Tables 1 and 2).

In open-loop control, the average cell model is used to precompute a temporal pattern of light stimulation so that cells follow a target behavior. This light pattern is then applied to all cells in the field of view (Fig. 3c and Supplementary Movie 3). In closed-loop population-based control, the average cell model and the average of the measured fluorescence of cells are used by classical state estimators and model predictive controllers to compute in real-time the appropriate light stimulation to drive the mean fluorescence to its target (Fig. 3d and Supplementary Movie 4). Finally, in closed-loop single-cell control, a stochastic model of gene expression and single-cell fluorescence measurements are used by advanced state estimators and controllers to compute in real-time the appropriate light stimulations to drive the fluorescence of each and every cell in the field of view to its target (Fig. 3e and Supplementary Movie 5). This control problem is quite challenging and needs to be solved for hundreds of cells in parallel. Advanced methods for numerical simulation and state estimation were essential (see Supplementary Note 5 and Supplementary Fig. 8).

Defining control performance as the time averaged deviation to target, we found that the single-cell control method leads to a modest reduction of error of the population-averaged fluorescence but to a drastic improvement of the average error of the single-cell fluorescence (Fig. 3f).

**Patterns of recombined yeast cells.** For our third application, we constructed a light-driven artificial recombination system in yeast and employed different light stimulation strategies to obtain various structures of recombined cells. We again used the EL222 optogenetic induction system but this time to drive the expression of the Cre recombinase. The Cre recombinase induces the expression of a fluorescent reporter, mCerulean, fused to a variant of the Far1 protein, Far1M, via an amplification step using the ATAF1 transcription factor (Fig. 4a, b). Far1 is the downstream effector of the mating pathway, and Far1M has been shown to arrest growth upon overexpression even in absence of mating factor[28]. This strain was designed to exhibit a growth arrest upon recombination as shown previously[29]. Firstly, we applied a ring-like recombination signal. More specifically, every cell that was in the designated zone at any moment throughout the experiment has been targeted for recombination (Supplementary Movie 6). As a result, we did obtain a ring-like pattern of recombined cells (Fig. 4c). Experimental and biological limitations can be revealed by the analysis of the tails of the distributions of the recombination readout (i.e., mCerulean fluorescence) within the cell populations (Fig. 4c). For example, we found that some cells have been erroneously targeted for recombination because of tracking issues, and that only a few cells have not shown the recombined phenotype at the end of the experiment despite having being effectively targeted for recombination (Fig. 4c and Supplementary Fig. 10).

Secondly, we tried to create islets of recombined cells. To this end, we dynamically searched for cells that were far from any previously-targeted cell, and targeted these cells for recombination. To maximize the chances that the chosen cells do recombine, we tracked each chosen cell and targeted it repeatedly with light stimulations (Supplementary Movie 7). Our strategy was effective in creating isolated micro-colonies of recombined

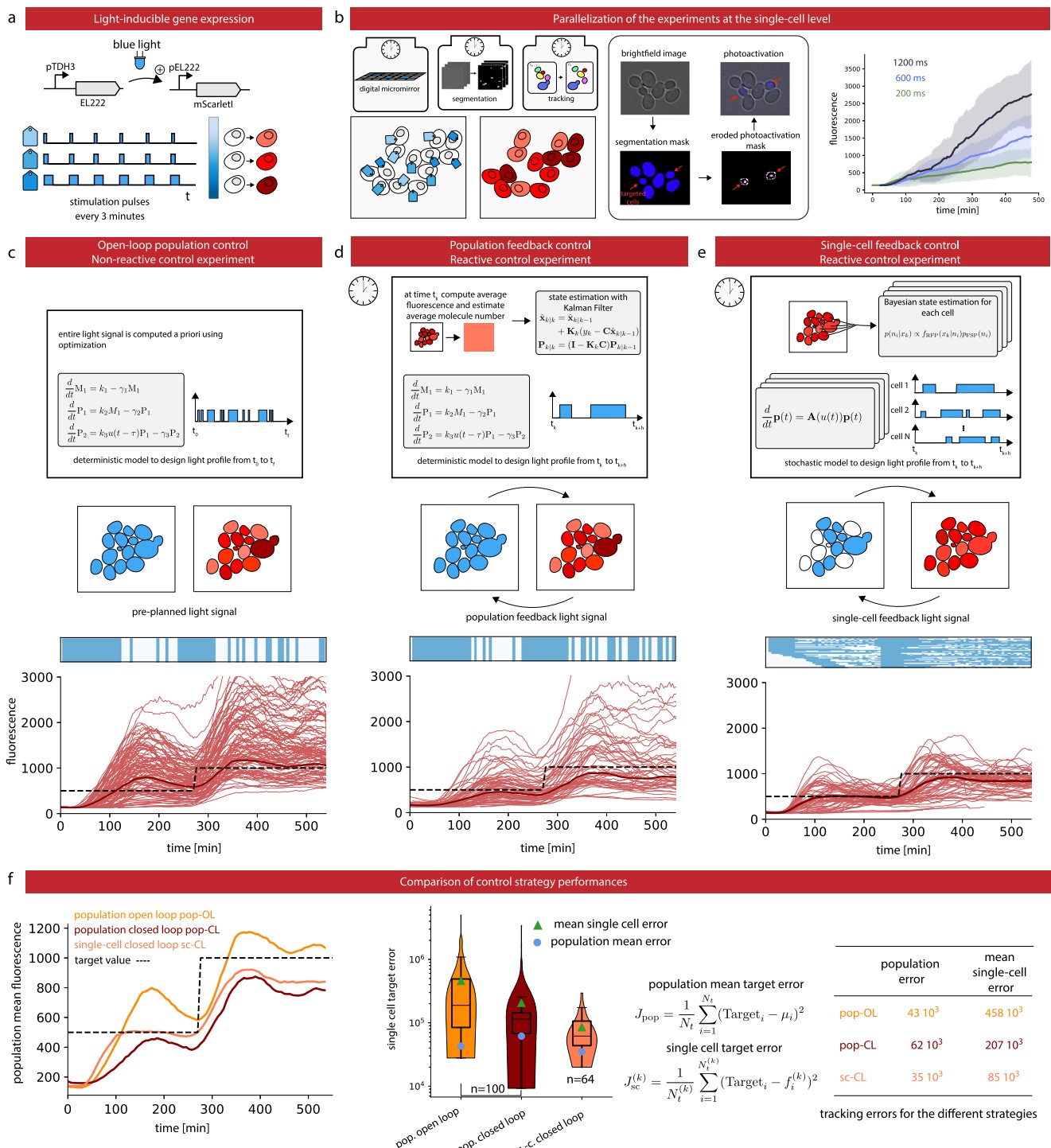

**Fig. 3 Control gene expression at the single cell level in yeast. a** The red fluorescent protein mScarletI is placed under the control of the light-responsive transcription factor EL222. **b** To efficiently characterize cell responses to light stimulations, cells in the field of view are partitioned in three groups, each group being stimulated with a different temporal profile. Bright-field images are segmented and cells are tracked. Then, based on their groups, cells are stimulated during the appropriate time with eroded masks (see Supplementary Fig. 5). Therefore, characterization experiments are run in parallel thanks to the DMD and our capacity to segment and track cells in real time. The temporal evolution of the mean mScarletI fluorescence of the cells in the three groups is shown with envelopes indicating one standard deviation. Single-cell trajectories and replicates are provided in Supplementary Fig. 6. **c** Open-loop control experiment in which a model of the response of the cell population is used to precompute a light stimulation profile that drives the cell population to the target behavior. The application of the light profile leads to significant deviations from the target of the individual cell trajectories. **d** Closed-loop control experiment in which the same model is used jointly with real-time observations of the population state to decide which light profile to apply to all cells, using a receding horizon strategy. **e** A stochastic model of individual cell response is used jointly with single-cell observations to decide which light profile to apply to each cell. **f** The different strategies have similar performances to drive the mean fluorescence to its target, but the single-cell feedback strategy is significantly better to drive individual cells to their target profiles. Box plots indicate the lower quartile, the median, and the upper quartile of the target error, with the whiskers corresponding to 1.5 interquartile ranges. Each control experiment was replicated two times.

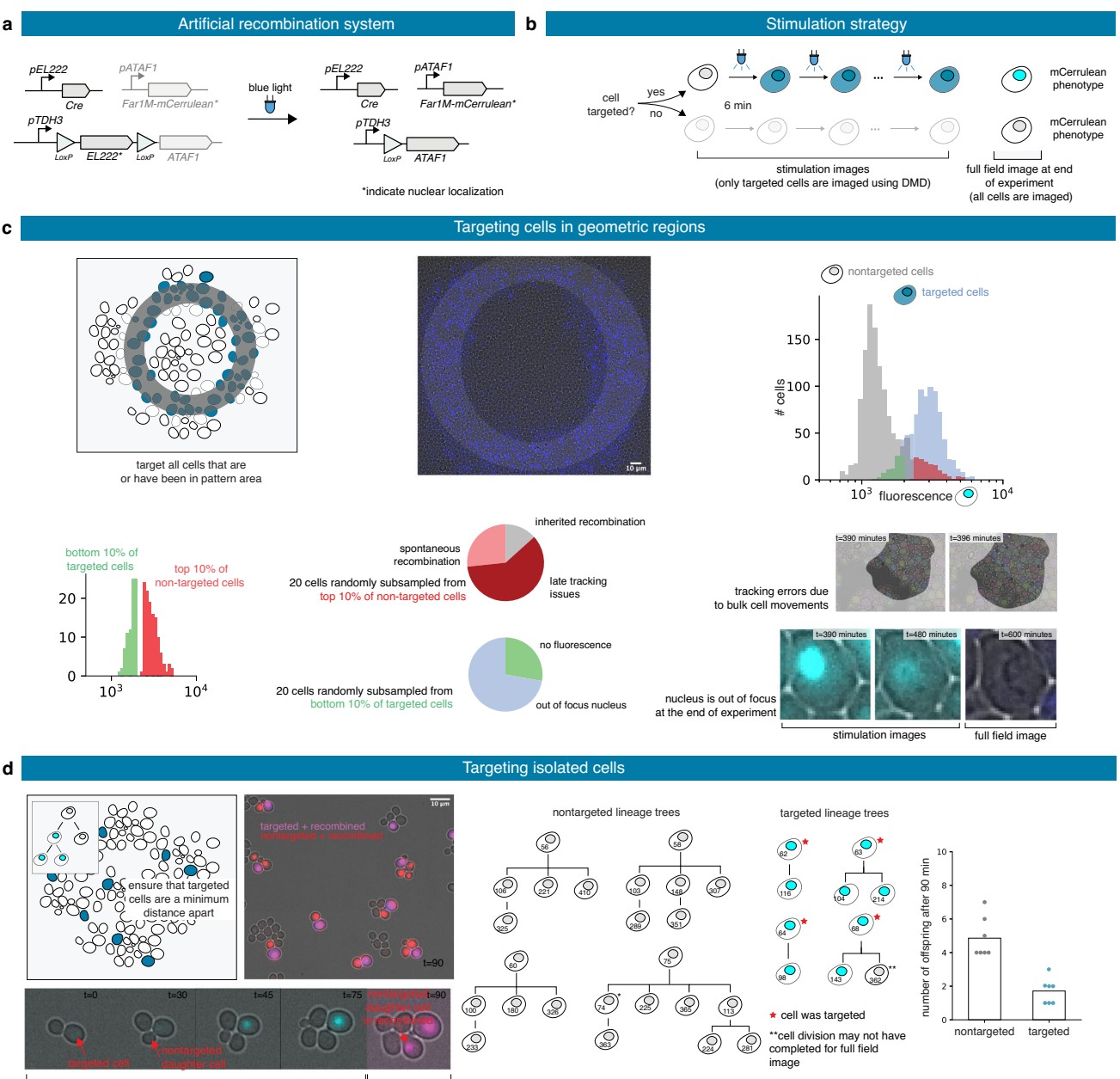

**Fig. 4 Patterns of recombined yeast cells. a** Upon light exposure, the Cre recombinase is expressed and triggers recombination, leading to the expression of ATAF1 and then of Far1M-mCerulean. Stars indicate nuclear localization of the protein. **b** Targeted cells are stimulated for 1 s every 6 min until the end of the experiment. Fluorescence levels emitted by targeted cells can be recorded. At the end of the experiment, all cells are imaged and a recombined or non-recombined phenotype is attributed. **c** A ring-like region in the field of view is selected at the beginning of the experiment and all cells entering the designated region at some time point are targeted for recombination. The distributions of the fluorescence levels of the targeted and non-targeted cells can be computed at the end of the experiment. The vast majority of cells present the expected phenotype and outliers can be further analyzed. **d** Cells are dynamically selected such that no target cells are close to each other. Cell lineages of targeted and non-targeted cells can be manually reconstructed and statistics can be extracted.

cells (Fig. 4d). Analysis of the lineage trees of targeted cells and non-targeted cells showed that recombined cells have a slow growth phenotype, even if growth arrest of recombined cells was not complete. The addition of a positive feedback loop on ATAF1 expression can lead to a stronger growth arrest, at the cost of a slightly higher level of spontaneous recombination[29]. Previous works demonstrating optogenetically-driven recombination use static masks for light targeting[25,29]. Obtaining single-cell resolution as demonstrated in Fig. 4d necessitates real-time image analysis and the use of reactive software.

## Discussion

We presented MicroMator together with one simple and two challenging applications. Altogether, these applications illustrate the genericity of MicroMator. Moreover, the latter two applications go beyond the state of the art and demonstrate how this software can help using automated microscopy platforms to their full potential. We demonstrated that protein expression can be controlled at the single cell level in dense fields of cells. This requires one to jointly address two challenges, namely obtaining sufficiently precise single-cell stimulations with DMDs and

segmenting and tracking cells with sufficient accuracy over extended durations. We also demonstrated that cell recombination can be triggered at the single-cell level, enabling single-cell resolution patterns. In comparison with Pycro-Manager, Micro-Mator uses the Micro-Manager GUI to create a main acquisition backbone for the experiment and reactive events are then used to enhance or even dynamically modify this initial plan. Events are created by default as separated threads and an extended logging system gathers messages from all modules that might be running in parallel. This structure provides robustness to real-time issues and facilitates error identification, two critical aspects for developing long and complex experiments.

MicroMator can be used to implement highly complex experiments. Yet, we foresee that reactiveness in microscopy will primarily be used to enhance and automate classical experiments. Examples of simple use cases abound: triggering autofocus only when needed, dynamically adjusting the imaging condition to the signal strength, identifying novel regions of interest, or following the course of experiments via easily accessible online services (e.g., warning messages sent on Discord), to provide but a few examples. Thanks to its modular nature and to its use of a simple but powerful event system, MicroMator capacities can be conveniently expanded to drive novel hardware or perform a wide range of analyses. MicroMator is a relatively simple software extension that significantly empowers laboratory equipment that is present in most quantitative biology laboratory worldwide.

## Methods

**Software**. MicroMator is an open-source software. It contains a core part and an extensible list of modules. The MicroMator core manages the user-specified events and also the metadata and logging system. The current list of modules includes a Microscope Controller module, two Image Analysis modules, a MPC module, and a Discord Bot module. The Microscope Controller module is an interface with the Python wrapper for Micro-Manager[17,18] pymmcore. The Image Analysis module SegMator uses the DeLTA deep learning method to segment yeast cells from bright-field images. It also uses an efficient algorithm for cell tracking. We also propose a second Image Analysis module, based on Cellpose. The MPC module implements state estimation and MPC routines for deterministic and stochastic systems, at either the population or single-cell level. The Discord Bot module uses a web app running on the microscope's computer and connected to the Discord communication system.

**Genetic constructions and strains**. For bacterial experiments, we used a *Corynebacterium glutamicum* strain carrying a plasmid expressing Wag31(Cgl2150) fused C-terminally to mNeonGreen, under control of *PgntK* promoter and constructed as in Sogues et al.[30]. For yeast experiments, all plasmids and strains were constructed using the *Yeast Tool Kit*, a modular cloning framework for yeast synthetic biology[31], the *S. cerevisiae* strain BY4741 (Euroscarf), and the EL222 optogenetic system[27]. The light responsive strain (IB44) harbors a constitutively expressed EL222 light-responsive transcription factor (NLS-VP16AD-EL222) and an EL222-responsive promoter (5xBS-CYC180pr) driving the expression of the mScarletI protein. The IB44 strain genotype is MATa his3Δ1 leu2Δ0::5xBS-CYC180pr-mScarletI-Leu2 met15Δ0 ura3Δ::NLS-VP16AD-EL222-URA3. The recombining strain (IB237) harbors a constitutively expressed EL222 light-responsive transcription factor (NLS-VP16AD-EL222) floxed between two LoxP sites that upon recombination expresses the ATAF1 transcription factor. This factor expresses (pATAF1_4x) in turn the mCerulean fluorescent protein fused to a constitutively active Far1 protein (FAR1M_mCerulean). Lastly, the strain also harbors the Cre recombinase placed under the control of an EL222-responsive promoter (5BS-Gal1pr). The IB237 strain genotype is MATa his3Δ1::pATAF1_4x-FAR1M_mCerulean-tDIT1-HIS3 leu2Δ::5BS-Gal1pr-CRE-tENO1-LEU2 met15Δ0 ura3Δ:: pTDH3-LoxP-NLS-VP16AD-EL222-tENO1-LoxP-ATAF1-tTDH1-URA3. Lastly, we also used the IB84 strain as a constitutive 3-color strain to characterize DMD precision. The genotype of this strain is MATa his3Δ1 leu2Δ0::pTDH3-mCerulean-tTDH1-pTDH3-NeonGreen-tTDH1-pTDH3-mScarlet-tTDH1-LEU2 met15Δ0 ura3Δ:: NLS-VP16AD-EL222-URA3. mCerulean, mNeonGreen, and mScarletI genes have been synthesized from IDT. We used sequences from Rizzo and Piston[32] for mCerulean and from Argüello-Miranda et al.[33] for mNeonGreen and mScarlet-I, and codon-optimized them for yeast using IDT codon optimizer software. Note that for mScarletI the sequence was modified to avoid a NotI restriction site by point mutation prior to optimization.

**Culture preparation**. Bacterial cells were grown in brain-heart infusion (BHI) for 6–8 h, then pelleted at $5200 \times g$ at room temperature and inoculated into CGXII media[34] supplemented with 4% sucrose and kanamycin (25 µg/mL) for overnight growth. The following day the culture was diluted to $OD_{600}$ of about 1 and grown for about 1.5 h to a required $OD_{600}$ of about 2. For each sample, 100 µL of culture were pelleted, washed with fresh media, and concentrated to an $OD_{600}$ of 3 for imaging. Yeast cells were grown at 30 °C in synthetic medium (SD) consisting of 2% glucose, low fluorescence yeast nitrogen base (Formedium CYN6510), and complete supplement mixture of amino acids and nucleotides (Formedium DCS0019). For each experiment, cells were grown overnight in SC media at 30 °C, then diluted 50 times and grown for 4–5 h before being loaded in microfluidic plates.

**Microscopy setup, agar pads, microfluidics, and imaging**. Images were taken under a Leica DMi8 inverted microscope (Leica Microsystems) with a ×63 oil-immersion objective (HC PL APO), an LTM200 V3 scanning stage, and an sCMOS camera Zyla 4.2 (ANDOR). Bright-field images were acquired using a 12 V LED light source from Leica Microsystems. Fluorescence images were acquired using a pE-4000 light source from CoolLED and the following filter cubes: EX:436/20 nm DM:455 nm EM:480/40 nm (CFP), EX:500/20 nm DM:515 nm EM:535/30 nm (YFP), and EX:546/10 nm DM:560 nm EM:585/40 nm (RHOD) from Leica Microsystems. Light stimulation was performed using the pE-4000 light source using the 435 nm filter, a light intensity of 20%, and the CFP filter cube. Spatially-resolved illuminations were obtained thanks to a digital mirror device (DMD) reflecting the light of a pE-4000 light source. We used a MOSAIC3 DMD from ANDOR. The device is used both for targeted fluorescence imaging and for optogenetic stimulations. For *C. glutamicum*, Nile Red (Sigma-Aldrich) was added to the culture (2 µg/ml final concentration) for membrane staining, just prior to placing cells on pads. We used 2% agarose pads prepared with the corresponding growth medium and covered with a glass coverslip. A hole was cut into the pad to enable oxygen supply required for growth. To grow yeast cells in monolayers, a CellASIC ONIX2 system (Merck) was used together with the Y04C CellASIC microfluidic plates. Media flow was maintained by a 7.5 kPa pressure gradient. The media was the same as for pre-culture. The temperature was maintained at 30 °C by an opaque environmental box and a temperature controller 2000-2, both from PECON (Supplementary Fig. 9). The microscope was operated using MicroMator. The computer running the microscopy platform and MicroMator is equipped with a CPU made of 2 10-core processors (Intel Xeon E5-2640 V4, 2.4–3.4 Ghz), with a Nvidia Quadro M4000 GPU, and with 64 GB of RAM (DDR4 ECC).

**Image analysis**. *C. glutamicum* images have been segmented in real time with an analysis module using the generalist, deep-learning-based segmentation method Cellpose[24]. *S. cerevisiae* images have been segmented in real-time with an analysis module called SegMator using the deep-learning-based tool DeLTA[21].Cell tracking was also implemented in SegMator and has been solved using TrackPy[22]. Our yeast applications necessitate good segmentation and tracking performance. The neural network used by DeLTA has been trained offline and its online performance is documented in the Supplementary Note 3. In all cases, the fluorescence of a cell is defined as the mean pixel intensity of the cell.

**Model predictive control of gene expression**. To compare single-cell and population control strategies, we developed stochastic and deterministic models of gene expression. Both have been calibrated with respect to the dataset presented in Fig. 3b and Supplementary Fig. 6. For population control, we used the deterministic model, assumed Gaussian measurement noise, and used a Kalman filter for state estimation. Each model assumes a deterministic delay between the time the light signal is applied and the time protein production is effective. For MPC, fluorescence measurements were taken every 6 min and we considered receding time horizons of 24 min. The controller explores the set of light stimulation profiles in which a 1000 ms light stimulation is either applied or not for each measurement time interval, and selects the profile minimizing mean square deviations. For tracking purposes, brightfield measurements were taken every 3 min. For single-cell control, we used the stochastic model and simulated the cell behavior using a finite state projection approximation. For each and every cell, state estimation is performed using a Bayesian approach which conditions the probability distribution for each cell on the most recent measurement, and light stimulation profiles are selected using the approach outlined above and the expected absolute deviation as selection criterion. More information is provided in Supplementary Note 5.

**Reporting summary**. Further information on research design is available in the Nature Research Reporting Summary linked to this article.

## Data availability
Raw and processed data for Supplementary Fig. 1 (tutorial) and Fig. 2, and for Figs. 3c–e, 4c–d and Supplementary Fig. 5 are freely available on zenodo repositories: https://doi.org/10.5281/zenodo.5761545 (23GB) and https://doi.org/10.5281/zenodo.4616659 (45GB). Source data is available as a Source Data file.

## Code availability

The MicroMator software, together with event definitions for representative experiments (Figs. 2c, 3e and Supplementary Fig. 5), can be found online: https://gitlab.inria.fr/InBio/Public/micromator. Data analysis code for experiments (Figs. 3e, 4d, and Supplementary Fig. 5), as well as a tutorial example (Supplementary Note 5) and a tutorial movie (Supplementary Movie 1), can be found at the same place.

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

## Acknowledgements

This work was supported by ANR grants CyberCircuits (ANR-18-CE91-0002), MEMIP (ANR-16-CE33-0018), and Cogex (ANR-16-CE12-0025), by the H2020 Fet-Open COSY-BIO grant (grant agreement no. 766840) and by the Inria IPL grant COSY. We acknowledge the support of the U.S. Department of Energy through the LANL/LDRD Program and the Center for Nonlinear Studies for this work. We thank Anne-Marie Wehenkel for her detailed comments.

## Author contributions

S.F. developed the MicroMator software and performed experiments. Z.F. wrote the image analysis and real-time control modules, and analyzed the data. A.F. performed experiments and helped develop the MicroMator software and analyze the data. C.A. developed strains and helped perform experiments. S.S.-C. developed strains. J.P. developed strains and helped with microscopy experiments. S.G. helped with software development. F.B. helped with software development and platform integration. J.R. helped with controller development. Z.F., F.B., J.R., and G.B. supervised the project. Z.F. and G.B. wrote the manuscript with input from all authors. A.F. and C.A. contributed equally to this work.

## Competing interests

The authors declare no competing interests.
