## [Peer Review File · Nature Communications]

Reviewers' Comments:

Reviewer #1:

Remarks to the Author:

The authors present a software for microscope control for reactive experiments stimulating cells on a single-cell level. I fully agree with the authors that such an approach is highly relevant for reproducibility of micro manipulation / microscopy experiments. The authors mention that the presented applications of their software goes beyond state of the art and I tend to agree. The demonstrated combination of theoretical modeling, microscope control and experimentation appears to me like a masterpiece of modern science and engineering. The authors share the software open source and provide data open access. That is greatly appreciated. I also would like to acknowledge the supplementary material which is very helpful to understand how the system works. All together an impressive piece of software development and very well documented. I do have three major comments and furthermore wrote down some minor aspects which I came across while reading. If the authors want to just ignore the one or the other minor comment that will be fine.

Major comments:

- * The manuscript describes a "simple software solution" (page 2, line 44) and I was expecting a software application with an end-user ready interface. I'm wondering if it's more reasonable to describe it as "software library". I have the impression that MicroMator is rather an application programming interface (API) separated in multiple modules. Am I right, MicroMator is best suited for programmers?
- * Maybe the authors could spend another 2-3 sentences in the introduction section explaining the experiments they are automating. For a reader from a different field it might be a bit hard to capture the complexity of the performed experiments.
- * In general I miss quantification of the software's performance. The authors mention "good accuracy" (Page 2, line 67) and "significantly diminished" (Supplementary page 5). I'm wondering if it's possible to enrich the manuscript with sound quantitative data in the context of segmentation and tracking quality.

Minor comments:

Page 1, line 15 and line 26: "enormous progress both in terms of accuracy and speed thanks to machine learning methods" and "image analysis has recently made a giant leap in terms of accuracy and rapidity thanks to deep learning methods." I'm not sure if I can agree. Machine / deep learning methods typically come with high computational costs and thus, can rather not be the origin of speedup. I would argue that image processing and machine/deep learning made progress in terms of speed because of development in graphics processing unit (GPU) technology.

Page 2, line 44: The authors mention "a simple software solution". Would it be possible to add a screenshot of the user interface to the supplementary material to demonstrate simplicity? Furthermore, a more detailed specification would be appreciated: Who is the target audience? Can life scientists without coding skills use the "simple software" or are programming skills on python expert level necessary?

Page 2, line 48-49. I'm not sure if I understand what "if message !update position=10 frame=last is received from Discord" means. Maybe the examples mentioned in this section would be suited for a table with more detailed explanation.

Page 2, line 60. I wasn't aware that "Discord" is a thing. Would it be possible to cite it, mention its vendor and/or write a sentence about what Discord is?

Page 2, line 67. The authors mention "good accuracy" and link to supplementary text, Figure and Movie. However, I didn't find any specification of the algorithms accuracy in the supplementary material. Was accuracy of the algorithm quantified in this work?

Page 3, line 85: Would it be possible to provide a reference behind "EL222 optogenetic system" and "mScarletI"?

Page 5, line 130: Would it be possible to provide a reference for "mCerulean".

Supplementary material

Page 2, second section: "... which are broken..." In my opinion "broken" is the wrong word here. Maybe "separated" is better?

Page 3, second section: Please cite the pandas library, e.g.: Data structures for statistical computing in python, McKinney, Proceedings of the 9th Python in Science Conference, Volume 445, 2010.

Page 3, third section: "A key advantage of deep learning based approaches is that they are computationally expensive to train..." I think this could be rephrased to minimize confusion.

Page 3, third section: "... which automatically apply random transformations of flip/shift/rotate..." the term for this is "image augmentation". Read more:
<https://journalofbigdata.springeropen.com/articles/10.1186/s40537-019-0197-0>

Page 3, third section: "Code and our pretrained model files are available on Gitlab." would it be possible to provide a link?

Page 5, first section: "significantly diminished, as shown in Fig. S9" This sounds like a statistical test was applied. Are there any quantitative details on tracking quality available? Furthermore, I'm not sure how Figure S9 (page 19) is related to tracking quality.

Page 12, last paragraph: A citation for "Krusalov subspace methods" would be appreciated.

Page 15, second section: The Kalman filter could be cited, e.g. Kalman, R. E. (1960). "A New Approach to Linear Filtering and Prediction Problems". Journal of Basic Engineering. 82: 35-45. doi:10.1115/1.3662552. S2CID 1242324.

Sincerely,
Robert Haase
robert.haase@tu-dresden.de

Reviewer #2:

Remarks to the Author:

In this article, Fox et al. introduce MicroMator, a microscope control software that can be used to perform reactive imaging experiments. Smart microscopy is a very hot topic area, and the development proposed by the authors is very welcomed. However, the paper in its current version is very challenging to review and reads more like an advertisement than a scientific paper. The claims also appear vastly overstated as it appears that the software in its current form can primarily be used to track yeast.

This is not helped by the fact that this reviewer could not test the software during the review period. I would encourage the authors to provide videos showcasing the software in action to demonstrate better what can be done using their software (i.e. screen recording). I listed below the comment that came to mind while reviewing the manuscript.

1. What is MicroMator?

Reading the text, this reviewer feels that the authors are dancing around what MicroMator is. Is MicroMator a standalone software? This reviewer thinks MicroMator is an extension for μ Manager since μ Manager is required to install MicroMator. Nothing wrong with this. However, this should be

clearly stated in the text and the abstract.

2. What is MicroMator for?

At the moment, the paper reads like an advertisement, and it is unclear what can be achieved using MicroMator. It is essential to list the strengths and current limitations of the software so that readers can make an informed decision.

The documentation provided with MicroMator states that "image analysis software for segmentation and tracking of yeast".

Can MicroMator be used for another purpose? This is not an issue, but it should be clearly stated in the text and abstract.

MicroMator relies on SegMator to analyse the acquired images in real-time and inform on how to proceed with the rest of the acquisition.

Can deLTA included in SegMator be trained directly in MicroMator? or does deLTA need to be trained outside of MicroMator?

What post-processing steps are used to achieve instance segmentation? Or is the tracking performed on binary masks directly? Can the users assess the quality of segmentation and tracking (ie using quality metrics)?

Delta has been mainly used to track bacteria. Here the authors demonstrate their automated pipeline on yeast. How will their software perform when following cells with more complex shapes?

Are other segmentation strategies implemented in SegMator?

Can the segmentation and tracking work on 3D images (i.e. Z stacks), or is it currently limited to 2D images?

The python tracking algorithm used is TrackPy. What is tracked here? The whole segmented yeast? or the centre of mass? Will the tracking work well on fast-moving cells?

How long does the feedback between acquisition and analysis take? What is the maximal acquisition frame rate? The author demonstrates one image every 3 minutes. Is it the maximal speed? How will this scale with the hardware running MicroMator?

Can other Deep Learning segmentation strategies used/ be implemented in MicroMator?

3. Who is MicroMator for?

MicroMator appears to have a GUI to control many features, but is python coding required to use it to its full extent? For example, are the users expected to write custom functions to trigger acquisition following a specific event?

From reading the documentation, the users need first to use μ Manager, then close it and open MicroMator to start an acquisition. The actual use process should clearly be described in the manuscript. For instance, Fig1 indicates that everything happens in MicroMator.

The material and method is minimal and is not sufficient to reproduce the experiments described here. For instance, no information is provided at all on how the images were analysed or processed. How was DeLTA trained? How many images were used for the training? What settings were used for TrackPy? How is the quality of segmentation and tracking assessed?

Minor comments:

I would recommend the authors fully annotate their supplementary video to understand what we are looking at. Highlighting the trigger, the detection and the reaction would be instrumental.

Reviewer #3:

Remarks to the Author:

In this paper, the authors describe their new software for reactive microscopy, MicroMator. Their key claims that current tools for reactive microscopy experiments are generally limited: users generally produce in-house solutions, with pre-existing solutions requiring extensive computational expertise. In contrast, their software claims to offer a simpler workflow by breaking down events into "Triggers" and "Effects", specific measurements or observations that result in specific responses. Moreover, it offers some interesting features like the ability for remote interaction - while I'm not sure how useful this is in practice, the promise of being able to walk away from an experiment and manage it instead through Discord is very cool. The main manuscript mainly focuses on demonstrating two use cases of reactive single cell applications - these both look reasonable to me in my limited expertise as a primarily computational person. I will instead focus my review on the technical components.

Overall, my main concern would be if the image analysis modules in this software are general or not, and the difficulty in creating custom modules. I'm currently under the impression that the moment the user wants to operate on images slightly different from the authors' set-up, or do tasks other than the arbitrary ones that scripts have been provided for, the software now requires extensive fine-tuning and coding efforts - which seems to contradict the authors' claims that their software is simpler to work with. It's possible that I'm missing a key piece of documentation or element of the GUI, and would be open to the authors pointing this out if so.

1. As this software claims to be general, and not an in-house solution, it would be critical to demonstrate that the segmentation and tracking modules are capable of generalizing to images from different groups. The authors fine-tune a pre-trained U-Net model with 50 images from their own lab, and while this might result in a model that works fine in-house for their particular microscopy set-up, I'm not sure if the model is actually general enough to function out-of-the-box on images from different groups without this fine-tuning procedure. I would like to see both quantitative and qualitative results for a variety of other yeast cell images - for example, the benchmark by the Yeast Image Toolkit. It would also be additionally useful if the authors could show generalization to cell markers outside of brightfield images (e.g. cytoplasmic fluorescent images). If it does turn out that annotating images for fine-tuning a model is essential to the performance of the segmentation module, this is a serious limitation that needs to be discussed, since it imposes time and hardware (e.g. a GPU for training) demands, or at least it requires the users to custom-write a module to an alternative segmentation method.

2. Just due to the nature of how diverse these experiments are going to be, it is very likely that users will have to write custom modules. Unfortunately, writing custom modules seems to require highly in-depth knowledge of the code. While the documentation is good at giving a general overview, my main issue is that even accessing and changing basic variables seems to require an exact knowledge of where everything is stored and how they are named. For example, to change the exposure value, I have to know to access the "protocol" object under the micromator core class and know what exact variables to access. Is there any way the authors can streamline or simplify this process (e.g. by having the GUI having a range of standard effects/triggers that the users can parameterize, or by providing better-commented and documented examples that can serve as wrappers - e.g. looping through all cells and calculating X should be a pretty standard thing)? At very least, the in-depth specifics of how parameters are stored and laid out, and what objects control what specifically should be better documented.

Reviewer #4:

Remarks to the Author:

While conceptually the reactive control system is brilliant, and the presented data show considerable improvement in achieving target expression (minimizing error), there are some key points that if addressed would make the description of this system clearer.

1- As it can be inferred from fig 2b, the response time (from EL222 being activated to mScarletI being detected) is considerable. This raises the question on how such delay in induction/detection is computed in the feedback information. This is somehow addressed in the supplementary file, where the model is calibrated and validated. Yet, as this is key, it should be better addressed in that section. Particularly, this should be contextualized and compared with other kinetic studies involving EL222 (i.e Benzinger & Mustafa Khammash PMID: 30166548). In addition, according to published data mScarletI has a maturation half-life of 36 min (<https://doi.org/10.1371/journal.pone.0219886>). How this coincide with the kinetics the authors observe and particularly, with the apparent decay of signal ?

2- While Fig 2B is informative regarding how different duty cycles lead to different degrees of expression, other critical description of the system is missing, regarding the off kinetics: how fast does mScarletI expression levels come down once lights are turned off (and how they go up upon induction, see previous point). All of this is particularly relevant if the utilized reporter does not have a degradation tag, nor its mRNA has a destabilizing signal. This is not explicitly commented (once again, see i Benzinger & Mustafa Khammash PMID: 30166548).

3- Figure S6A: This inset, containing single cells measurements under defined Light ON:OFF regimes is quite informative on the dynamics of the system and how this can be controlled AND monitored at the single cell level. This dataset could be included as part of Figure 2 (in some way it appears to be more relevant than current 2B)

4- The time scale of the data acquisition (2B, C,D, E) is of 500 mins, which is several times above the typical doubling time of yeast cells (~100 mins). Therefore, it is expected that as monitoring each cell, many of them will be budding and diving in several occasions. The latter may create a transient decrease in signal (due to the immediate dilution of mScarletI, and even a prolonged decrease in signal light-activated EL222 is also diluted). Could the authors comment on whether cell division was accompanied with such a decrease in signals, how cell division was accounted for when monitoring single cells, and what happened regarding subsequent decisions (are new daughter cells not considered in further analyses IF expression deviated too much?, etc). While in supplementary material this is indirectly addressed ("When new cells were born, they were assigned a stimulation using our cell-sorter event, such that the population of cells in each bin was approximately even"), the supplementary text would be enriched by addressing these key issues.

5- Related to the latter point, by looking at video 1 it is clear that, indeed, cells are dividing, yet cells at the edge of the cell cluster are dividing more actively than the ones remaining at the center of the clusters. How does signal intensity vary in the different population/age of cells? Visual inspection of fluorescence (Movie 2) suggests that there may be such a difference, yet looking at Fig 2 or S5 it is not possible to distinguish whether that is the case.

6- Suppl info: "in which we randomly assigned cells in the field of view to receive 0, 200, 500, or 2000 ms of photostimulation"

Could the authors comment on the intensity of light that was applied, how this compares to other EL222 yeast experiences (i.e Zhao et al PMID: 29562237; Benzinger & Mustafa Khammash PMID: 30166548), and whether intensity can be tuned with MicroMator on individual cells (yet without compromising a given erosion i.e. 33%). Importantly, as in most publications a given light intensity is applied to a cell culture of a particular OD, herein light is directly applied to cells (therefore that should be somehow accounted for)

7- Fig S4E: Please make sure the data present in the median Off-target fluorescence is correct.

8- Fig S5: The basal fluoresce (0 ms of photostimulation) is rather high. Could this be due to background environmental light in the place where the microscope is and/or samples are handled?

Strictly speaking one would expect the target cells/samples to be in the dark unless they are actively photostimulated. Was this the case, or was background light levels coming from general ceiling lights, monitors, screens etc? (Figure s8)

9- Figure 3. A and B legends are missing in the figure (it appears as the figure is cropped in the upper region)

10- Movie S3 and S4: There seems to be two different cell populations, ones with high fluorescence and others with low signals. Could the authors comment on that?

11- Line 225: "This factor expresses (pATAF1_4x) in turn the mCerulean fluorescent protein fused to a constitutively active Far1 protein (FAR1M_mCerulean)"

a) Why did the authors express a Far1- mCerulean fusion, and not just mCerulean?

b) Moreover, why did the authors use a constitutively active Far1?

c) Whereas methods talks about Far1_mCerulean figure 3 describes it as "mCerulean-Far1M" (c-terminal fusion).

d) The fact that upon light, CRE is activated, and Far1(+mCerulean) is expressed, can easily explain the slow growth phenotype (see line 144: "Analysis of the lineage trees of targeted cells and non-targeted cells confirmed that recombined cells have a slow growth phenotype"). The authors present this as an almost anecdotal result of recombination, whereas it would be an expected end-result (as Far1 expression should lead to cell cycle arrest). This should be properly explained

e) What is also confusing is that if the authors decided on purpose to express Far1M, this should have led to a stronger cell cycle arrest, which is not so obvious (as cells continue to divide albeit slower)

12- Line 238 "Microscopy setup, microfluidics and imaging"

There is no description on the illumination conditions of the MicroMator experimentation room setup, and how "darkness" was achieved when samples (cells) were not being actively illuminated (safety red-lights)? This info is critical as:

a) the proof of concept figures are based on comparing not illuminated versus illuminated cells

b) there seems to be a clear background level of mCerulean, which at this point is not clear if it is due to leakiness of EL222 control, or ambient light.

c) could background EL222 activity (due to ambient light or EL222 leakiness) also lead to cells exhibiting recombination even in the absence of direct illumination?

13- The authors do not provide additional information on the reporter mScarletI:

a) where was it obtained from (addgene, reference etc),

b) is it yeast optimized ?

c) does it have a degron sequence?

d) why was it chosen over other well characterized destabilized reporters?

13- Fig 1. Inset 3. "if n_cells==12 " is double = correct?

We thank all reviewers for their positive comments on our work and for their suggestions to improve the article.

Below we provide detailed comments to their remarks. Because we have made extensive changes to the article, adding entire sections or entire paragraphs, we do not quote here the modified or added text. Instead, we refer to the reviewers' versions of the main text and supplementary material in which significant changes have been highlighted in red.

Reviewer #1

The authors present a software for microscope control for reactive experiments stimulating cells on a single-cell level. I fully agree with the authors that such an approach is highly relevant for reproducibility of micro manipulation / microscopy experiments. The authors mention that the presented applications of their software goes beyond state of the art and I tend to agree. The demonstrated combination of theoretical modeling, microscope control and experimentation appears to me like a masterpiece of modern science and engineering.

We thank the reviewer for his positive appreciation of our work.

The authors share the software open source and provide data open access. That is greatly appreciated. I also would like to acknowledge the supplementary material which is very helpful to understand how the system works. All together an impressive piece of software development and very well documented.

We thank the reviewer for his positive assessment of our contributions.

I do have three major comments and furthermore wrote down some minor aspects which I came across while reading. If the authors want to just ignore the one or the other minor comment that will be fine.

Major comments:

The manuscript describes a "simple software solution" (page 2, line 44) and I was expecting a software application with an end-user ready interface. I'm wondering if it's more reasonable to describe it as "software library". I have the impression that MicroMator is rather an application programming interface (API) separated in multiple modules. Am I right, MicroMator is best suited for programmers?

MicroMator is primarily a command line software that reads user-defined configuration files that define the reactive experiments. A simple graphical user interface is provided as an alternative to writing command line calls. It is notably used to define which configuration files and which event definition files should be used.

One should also clarify that before running a MicroMator-driven experiment, one uses μ Manager and its graphical user interface to define experiment specific-configuration files that will be subsequently used by MicroMator. This notably includes the positions of interest in the sample and the imaging backbone of the experiment (a default image acquisition sequence with which MicroMator events will interact).

The different steps requested to launch a MicroMator experiment, including its initialization should be greatly clarified thanks to the video we made on a typical use scenario, as nicely suggested by another reviewer.

The core of the software effectively implementing the reactivity is small. This is why we qualified our solution as simple. Yet to perform challenging reactive experiments, one might use several modules that extend the software core. We already provide an extensive set of modules. They can be interfaces with other software or more complex pieces of code performing various real-time tasks. The list of modules can naturally be extended by the user. To ease its use, we also provide a list of predefined events of generic interest. They can be used as they are or they can serve as a basis for the definition of other events that are adapted to the specificities of the experiment or of the microscopy/microfluidic setup. Therefore, MicroMator is suited for users with basic knowledge of programming in Python, but does not require users to have extensive programming skills. Because the word “simple” can be understood in many different ways, we have chosen to remove it from the text to avoid ambiguities.

Maybe the authors could spend another 2-3 sentences in the introduction section explaining the experiments they are automating. For a reader from a different field it might be a bit hard to capture the complexity of the performed experiments.

We thank the review for this comment. We realized that the target community (biophysicists and quantitative biologists primarily) might not be fully aware of the enabling potential of reactive microscopy. We therefore provided a range of potential use cases in the first paragraph of the introduction.

In general I miss quantification of the software’s performance. The authors mention “good accuracy” (Page 2, line 67) and “significantly diminished” (Supplementary page 5). I’m wondering if it’s possible to enrich the manuscript with sound quantitative data in the context of segmentation and tracking quality.

First of all, we would like to clarify that we are not discussing here the accuracy of MicroMator but of SegMator, one image analysis module we implemented in MicroMator and used for our yeast applications with single-cell tracking. SegMator is based on DeLTA that itself uses the deep-learning U-Net architecture. Other image analysis tools might be used for other applications. For another case study, on *Corynebacteria* growing on agar pads (Figure 2), we developed another image analysis module based on Cellpose, another deep learning image analysis tool.

After training, the accuracy for yeast real-time image segmentation of DeLTA in SegMator is relatively good. This is now documented in the Supplementary Note 3. We found that our trained U-Net accurately segmented 1639 out of 1712 cells, spread across four images. The segmentation errors were dominated by false negatives (77 cells out of 1712), i.e. cells that were “missed” by the algorithm. Comparatively, we found much fewer false positives (only 5 cells out of 1712), corresponding to erroneous segmentation of non-cell pixels. We also found that most cells were accurately tracked. Analyzing 58 single-cell traces, most of them present during the entire duration of the experiment, we found that cells were correctly tracked in all but 4 cases. We added references in the main text to the Supplementary Note 3 and Supplementary Figures S2 and S3.

As mentioned above, we have added to the resubmitted version of the manuscript an experiment where we aim to quantify the mean intracellular fluorescence of *Corynebacterium glutamicum* cells growing on agar pads and stained with red Nile. For these experiments, we used the Cellpose algorithm [Stringer et al., 2021, Nat Methods] to segment cells without training on our specific

images. We added a new section and a new figure (Fig 2) to the main text, and also added information to the supplementary text in Supplementary Note 3 and Figure S4.

Minor comments:

Page 1, line 15 and line 26: “enormous progress both in terms of accuracy and speed thanks to machine learning methods” and “image analysis has recently made a giant leap in terms of accuracy and rapidity thanks to deep learning methods.” I’m not sure if I can agree. Machine / deep learning methods typically come with high computational costs and thus, can rather not be the origin of speedup. I would argue that image processing and machine/deep learning made progress in terms of speed because of development in graphics processing unit (GPU) technology.

It is important here to distinguish the training phase of the network and the simple use of the trained network (inference). The learning phase is computationally intensive and the use of GPU is essential at this iterative stage. However, the analysis of a new image by a trained network simply amounts to a forward pass in the network and uses only a modest amount of resources. As shown in Figure 1 below, segmentation takes a marginal amount of time in our applications. Standard imaging methods such as active contours for example necessitate an optimization phase on each novel image. Therefore, regarding the specific context of real-time image analysis, it seems to us that the enabling feature is to be able to pretrain the network off line and “only” perform online an inference pass on novel images.

In order to also recognize the importance of hardware improvements, we have modified our sentences to read “enormous progress both in terms of accuracy and speed thanks to machine learning methods and improved computational resources.” and “image analysis has recently made a giant leap in terms of accuracy and rapidity thanks to deep learning methods and software which leverage modern computational resources”

Page 2, line 44: The authors mention “a simple software solution”. Would it be possible to add a screenshot of the user interface to the supplementary material to demonstrate simplicity? Furthermore, a more detailed specification would be appreciated: Who is the target audience? Can life scientists without coding skills use the “simple software” or are programming skills on python expert level necessary?

We thank the reviewer for these comments and have added a new video (Supplementary Movie 1) that shows how the software can be used to implement a simple reactive experiment.

We now mention in the introduction a rather large range of potential applications and we have also expanded the set of case studies we provide. These additions should help clarify that our target audience is essentially made of biophysicists and quantitative biologists. Moreover, we also make it clear now that MicroMator is suited for users with basic knowledge of programming in Python, but does not require users to have extensive programming skills (See answer to major comment 1 too).

Page 2, line 48-49. I’m not sure if I understand what “if message !update position=10 frame=last is received from Discord” means. Maybe the examples mentioned in this section would be suited for a table with more detailed explanation.

Our Discord example was indeed convoluted. We replaced it by two examples with non-ambiguous meaning: “Warning: focus lost” and “Ending acquisition”.

Page 2, line 60. I wasn't aware that "Discord" is a thing. Would it be possible to cite it, mention its vendor and/or write a sentence about what Discord is?

Discord is an instant messaging software (and much more; video/images notably). It is owned by an independent company, Discord Inc. The service has more than 350 million users worldwide. We clarified in the text that Discord can be used for instant messages.

Page 2, line 67. The authors mention "good accuracy" and link to supplementary text, Figure and Movie. However, I didn't find any specification of the algorithms accuracy in the supplementary material. Was accuracy of the algorithm quantified in this work?

References to Figures S2 and S3 of the supplementary material were obviously missing in the main text. We have also refactored the supplementary material to separate the description of the MicroMator software itself (Supplementary Note 1) from the description of the performance of the image analysis modules we developed for our applications (Supplementary Note 3).

Page 3, line 85: Would it be possible to provide a reference behind "EL222 optogenetic system" and "mScarletI"?

Page 5, line 130: Would it be possible to provide a reference for "mCerulean".

We have updated the references to the EL222 optogenetic system, mScarletI and mCerulean at the prescribed locations.

Supplementary material

Page 2, second section: "... which are broken..." In my opinion "broken" is the wrong word here. Maybe "separated" is better?

Indeed. We have reworded this sentence for clarity.

Page 3, second section: Please cite the pandas library, e.g.: Data structures for statistical computing in python, McKinney, Proceedings of the 9th Python in Science Conference, Volume 445, 2010.

We have added this reference.

Page 3, third section: "A key advantage of deep learning based approaches is that they are computationally expensive to train..." I think this could be rephrased to minimize confusion.

We thank the reviewer for this comment and have reworded this sentence.

Page 3, third section: "... which automatically apply random transformations of flip/shift/rotate..." the term for this is "image augmentation". Read more:

<https://journalofbigdata.springeropen.com/articles/10.1186/s40537-019-0197-0>

We added an explicit reference to image augmentation.

Page 3, third section: "Code and our pretrained model files are available on Gitlab." would it be possible to provide a link?

We added a link.

Page 5, first section: "significantly diminished, as shown in Fig. S9" This sounds like a statistical test was applied. Are there any quantitative details on tracking quality available? Furthermore, I'm not sure how Figure S9 (page 19) is related to tracking quality.

The reference to Figure S9 was a mistake. We made the connection with data provided in Figure 4C in the main text.

Page 12, last paragraph: A citation for “Krulov subspace methods” would be appreciated.

Page 15, second section: The Kalman filter could be cited, e.g. Kalman, R. E. (1960). “A New Approach to Linear Filtering and Prediction Problems”. *Journal of Basic Engineering*. 82: 35–45. doi:10.1115/1.3662552. S2CID 1242324.

We added these two references.

Sincerely,

Robert Haase

robert.haase@tu-dresden.de

Reviewer #2

In this article, Fox et al. introduce MicroMator, a microscope control software that can be used to perform reactive imaging experiments. Smart microscopy is a very hot topic area, and the development proposed by the authors is very welcomed.

We thank the reviewer for their positive appreciation of our work.

However, the paper in its current version is very challenging to review and reads more like an advertisement than a scientific paper. The claims also appear vastly overstated as it appears that the software in its current form can primarily be used to track yeast.

MicroMator is a generic software for facilitating the development of reactive microscopy experiments. However, in our initial submission, we only provided examples of highly challenging experiments that all requested single-cell tracking of yeast (actually, we also provided a very simple example of dynamic adaptation of fluorescence illumination in the supplementary material, but its visibility was quite poor since it was not cited from the main text). Although it was probably appropriate to demonstrate the potential of reactive microscopy in general and of MicroMator in particular, as appreciated by several reviewers, we also realize thanks to the feedbacks of this and other reviewers, that our application choices strongly conveyed the impression that MicroMator is tuned for yeast microscopy experiments.

We implemented three changes that hopefully address this issue. Firstly, we now provide in the introduction a list of potential applications of reactive microscopy showing the diversity of the questions one can address. Secondly, we explicitly discuss in the software description paragraph the benefits of using MicroMator in comparison with custom made solutions that would need to combine several existing, disconnected tool (see also answers to “What is MicroMator for”). Thirdly, we now provide a third case-study using different biological material (bacteria, *Corynebacteria*), imaged in a different microscopy setup (agar pads), and analyzed using a different image analysis tool (Cellpose). In this problem, the red fluorescent dye used to stain the cells migrates and diffuses across the agar pad, leading to a fluctuating and decreasing fluorescent signal. To tackle this issue, we used MicroMator to adapt the exposure times of red channel acquisition at every time points to achieve a constant desired fluorescence value. We also noted that this contributed to improve the signal-to-background ratio. A novel section (“Real-time exposure adjustment in *Corynebacterium*”) and figure (Figure 2) has been added in the main text,

with corresponding extensions in the Material and Methods section and in the supplementary material.

This is not helped by the fact that this reviewer could not test the software during the review period. I would encourage the authors to provide videos showcasing the software in action to demonstrate better what can be done using their software (i.e. screen recording). I listed below the comment that came to mind while reviewing the manuscript.

The software is open and is available on gitlab at <https://gitlab.inria.fr/InBio/Public/micromator>. Yet, due to its nature, we understand that it is not easy to test the software for a rapid assessment. Therefore, we followed the reviewer's suggestion and made a video capture of the different steps needed to run an experiment. It is now Supplementary Movie 1. We thank the reviewer for their suggestion. We have also made additional references to the software documentation available on the gitlab repository and provided code snippets in the supplementary information to demonstrate the simplicity of Trigger and Event design.

1. What is MicroMator?

Reading the text, this reviewer feels that the authors are dancing around what MicroMator is. Is MicroMator a standalone software? This reviewer thinks MicroMator is an extension for μ Manager since μ Manager is required to install MicroMator. Nothing wrong with this. However, this should be clearly stated in the text and the abstract.

MicroMator is primarily a command line software that reads configuration files that define the reactive experiments. It is perfectly true that MicroMator is tightly related to μ Manager. Actually, MicroMator makes use of μ Manager in two different ways. First, as said in an answer to a comment by Reviewer1, we use μ Manager and its GUI prior to running a MicroMator-driven experiment to define experiment-specific configuration files. This notably includes the positions of interest in the sample and the imaging backbone of the experiment. This imaging backbone is a default image acquisition sequence with which MicroMator events will interact. Second, during an experiment, MicroMator uses the Python-based API of μ Manager pymmcore to drive microscope-specific operations. This API is used through the Microscope Module we provide. We do not hide the connection between μ Manager and MicroMator -even the name of our tool highlights the connection- and we rather see this as a strength given the large community of users.

We realized that the nature of MicroMator was indeed not sufficiently described in the main text and we have added a full paragraph in the MicroMator Software description that clarifies this. We have also improved Fig 1A to make appear the required configuration files (on the left) and added a video demonstrating the initialization steps of a MicroMator-driven experiment (Supplementary Movie 1). We thank the reviewer for their comment.

2. What is MicroMator for?

At the moment, the paper reads like an advertisement, and it is unclear what can be achieved using MicroMator. It is essential to list the strengths and current limitation of the software so that readers can make an informed decision.

The range of potential use cases of reactive microscopy has now been depicted by a list of illustrative examples added in the introduction. Moreover, we now propose three case studies, one making a relatively simple use of reactivity and two showcasing advanced use. For example, we show that using MicroMator, one can do image segmentation, cell tracking, model predictive control using stochastic models, parallelized at the single cell level, and single-cell optogenetic actuation. All these steps can be done in real-time and allow for the control of gene expression in individual cells.

Strictly speaking, with the provided code, the user can only implement experiments that reuse some of the predefined events we implemented and that use modules that we already wrote. However, the intended use is naturally that the user adapts the definition of events according to its needs based on the examples we provide and/or tailors or expands the lists of modules, mimicking the modules we wrote. Although some additional programming is involved, we would like to stress that extending a few portions of code following a predefined architecture and having diverse examples to take inspiration from, or writing a complete reactive microscopy program from scratch are two tasks of very different complexities that involve very different programming skills. This holds for the initial development of the software and even more so for its debugging, maintenance and reuse. We clarified this point in the main text.

On the technical aspect, one limitation of MicroMator is that it does not verify the satisfaction of real-time constraints. Triggered events are queued and treated in order. If the user requires that an imaging routine that lasts 4 minutes is triggered every 3 minutes, delays will accumulate. So, real-time issues might happen. However, we provide an extensive logging system to help troubleshoot potential issues. We now mention this limitation in the description of the MicroMator Core part of the software description (Supplementary Text 1).

The documentation provided with MicroMator states that “image analysis software for segmentation and tracking of yeast”. Can MicroMator be used for another purpose? This is not an issue, but it should be clearly stated in the text and abstract. MicroMator relies on SegMator to analyse the acquired images in real-time and inform on how to proceed with the rest of the acquisition.

SegMator is indeed “image analysis software for segmentation and tracking of yeast” as its documentation says. It is based on the popular DeLTA software, that itself is using the U-Net architecture. It was essential to demonstrate how image analysis tools can be used as toolboxes with MicroMator. However, it is just one image analysis tool. Others can be used with MicroMator. In the Supplementary Material (Writing your own Modules section) and in the MicroMator documentation, we illustrated how to develop an additional module by means of the development of a very simple image analysis module based on numpy. It simply computes the average pixel value in an image and has been used in the toy example described in Supplementary Note 2.

To answer this and other reviewers’ comments, we have now developed another application that uses another image analysis tool, Cellpose. The fact that several image analysis modules, based on different tools, are used in examples in the main text should help clarifying the generic aspect of MicroMator.

Can deLTA included in SegMator be trained directly in MicroMator? or does deLTA need to be trained outside of MicroMator?

The image analysis tools that are used by MicroMator need to satisfy the requirements of the specific task performed, notably in terms of accuracy, computational time, and memory. Our objectives with MicroMator is not to offer possibilities to train image analysis tools offline. This has to be done separately. We realized that the fact that the training steps done for DeLTA in SegMator were described in the “software description” part in the Supplementary Material was confusing. To address this issue, we now have two distinct Supplementary Notes, one on software description (Supp Note 1) and one on image analysis tools (Supp Note 3).

Actually, Segmator can be used as a standalone tool or as a MicroMator image analysis module. Within SegMator, we propose Python code to train DeLTA using a napari plugin to annotate one’s own images (file `seg_trainer.py` in the SegMator repository). However, as the image analysis pipelines are not the emphasis of our work here and to limit the risks of confusion between SegMator and MicroMator tools, we have chosen to provide only a limited visibility to this (nice) possibility.

What post-processing steps are used to achieve instance segmentation? Or is the tracking performed on binary masks directly? Can the users assess the quality of segmentation and tracking (ie using quality metrics)?

In SegMator, the tracking is performed directly on the binary masks. A key part of U-Net training is to use pixel-wise weight maps which emphasize the barriers between cells, thus preventing the common issue of multiple cells bleeding together. This alleviates the need to use watershed or other algorithms in a post-processing step. And indeed, we found that using weight maps effectively minimizes the bleeding-together of multiple cells, thus enabling us to track cells directly using the center of mass of the individual binarized masks. We added in the Cell Tracking section of the Supplementary Text that the tracking is done directly on the binary masks. However, we did not develop more this aspect since it is specific to U-Net in this case and not closely connected to MicroMator.

We did quantify the accuracy of image analysis with SegMator. These results were presented in Supplementary Figure 2. However, they were not referred to in the Main Text. Concretely, as already mentioned above, we found that our trained U-Net accurately segmented 1639 out of 1712 cells, spread across four images (Fig S3A). The segmentation errors were dominated by false negatives (77 cells out of 1712), i.e. cells that were “missed” by the algorithm. Comparatively, we found much fewer to false positives (only 5 cells out of 1712), corresponding to erroneous segmentation of non-cell pixels. We also found that most cells were accurately tracked. Analyzing 58 single-cell traces, most of them present during the entire duration of the experiment, we found that cells were correctly tracked in all but 4 cases. These results are shown in Figure S3B.

Delta has been mainly used to track bacteria. Here the authors demonstrate their automated pipeline on yeast. How will their software perform when following cells with more complex shapes?

Each specific application generally necessitates adaptation of the chosen image analysis tool to the specificities of the experimental setup (eg, brightfield or phase contrast imaging) and of the biological material (yeast, bacteria, etc). In this case, we found that DeLTA could be retrained to segment yeast with a good accuracy. This could give us hope that it could still perform well on other, more complex shapes, but no general guarantee can be provided of course. In any case, if

one specific image analysis tool is not appropriate for a particular task, other tools can be used with MicroMator, as illustrated with our additional case study that uses Cellpose.

Are other segmentation strategies implemented in SegMator?

We do not plan to extend the capacities of SegMator. We intend to expand the imaging capabilities of MicroMator by offering more imaging modules that make use of various existing tools. We have already started to test OmniPose for example.

Can the segmentation and tracking work on 3D images (i.e. Z stacks), or is it currently limited to 2D images?

For our applications, we have only used 2D images so far. However, we have now integrated the possibility to use the Cellpose algorithm, which has been used to segment 3D images. Given that TrackPy works for 2D and 3D data, it is in principle possible to work with 3D images in MicroMator using Python/Cellpose/TrackPy. We note however that imaging extensive Z-stacks and analyzing all images might be time consuming and might conflict with the real time requirement of some reactive applications.

The python tracking algorithm used is TrackPy. What is tracked here? The whole segmented yeast? or the centre of mass? Will the tracking works well on fast-moving cells?

As mentioned above, cell tracking is performed on the centroid or center of mass of the segmented yeast cells. While we haven't used the tracking software in 3D or for fast moving cells, TrackPy was developed for particle tracking, and has been used extensively for fast moving objects in three dimensions. Naturally, imaging fast-moving objects, relatively to the imaging rate, is challenging. For example, we found that if we image yeast every 6 minutes instead of every 3 minutes, our ability to track individuals was diminished in dense regions.

How long does the feedback between acquisition and analysis take? What is the maximal acquisition frame rate? The author demonstrates one image every 3 minutes. Is it the maximal speed? How will this scale with the hardware running MicroMator?

We documented the typical analysis time for our setup on Fig 1 (Below). As it can be seen, the full analysis of an image with up to 3000 cells always takes less than 25 seconds, mainly spent in disk operations and basic image computations (ie excluding segmentation). Note that these operations have not been optimized.

Fig 1. Computational time required for image analysis. The time needed for image segmentation, for cell tracking, for the computation of the fluorescence values of cells, and for disk operations is provided as a function of the number of cells present in the field of view. The data is based on the images collected for the control experiments presented in Fig 3 C, D and E in the main text.

For real-time control experiments, one should add to this the time needed for state estimation and model predictive control. The most time-constrained case we had was the single-cell control experiment of Fig 3E. We ran a stochastic state estimator (using Bayesian estimation) and solved a stochastic model predictive control problem (using a finite state projection based approach) for each cell. Again, the computational workload was the most intense towards the end of the experiment when the cells were the more numerous in the field of view. Then, this step took up to 30 seconds, using multithreading and code optimization. The computer running the microscopy platform and MicroMator is equipped with 2 10-core Xeon E5-2640 V4 processors, and an Nvidia Quadro M4000 GPU. We added the specifications of the computer we used in the method section.

Can other Deep Learning segmentation strategies used/ be implemented in MicroMator?

Yes. We have implemented an image analysis module using Cellpose for example. The corresponding Image Analysis module has been added to the git repository.

3. Who is MicroMator for?

MicroMator appears to have a GUI to control many features, but is python coding required to use it to its full extend? For example, are the users expected to write custom functions to trigger acquisition following a specific event?

From reading the documentation, the users need first to use μ Manager, then close it and open MicroMator to start an acquisition. The actual use process should clearly be described in the manuscript. For instance, Fig1 indicates that everything happens in MicroMator.

As mentioned previously, we realized that we were not sufficiently explicit on how to use MicroMator and on the target users of MicroMator. We thank the reviewer for drawing our attention on these facts.

We made a video capture of the initialization steps to run experiments with MicroMator (now Supplementary Movie 1). We also clarified the main steps in a novel paragraph (the second paragraph) of the MicroMator Software section in the main text.

We have also modified Figure 1 to represent explicitly that μ manager configuration files need to be provided as an input to the program, and expanded the caption of the figure to mention that μ manager is used beforehand to provide configuration files.

The material and method is minimal and is not sufficient to reproduce the experiments described here. For instance, no information is provided at all on how the images were analysed or processed. How was DeITa trained? How many images were used for the training? What settings were used for TrackPy? How is the quality of segmentation and tracking assessed?

We thank the reviewer for this comment. We expanded the Material and Methods section, notably adding a subsection on image analysis. We also expanded the Supplementary text on DeITa training and TrackPy configuration (see Supplementary Note 3).

Minor comments

I would recommend the authors fully annotate their supplementary video to understand what we are looking at. Highlighting the trigger, the detection and the reaction would be instrumental.

We have improved the annotations of all the videos.

Reviewer #3

In this paper, the authors describe their new software for reactive microscopy, MicroMator. Their key claims that current tools for reactive microscopy experiments are generally limited: users generally produce in-house solutions, with pre-existing solutions requiring extensive computational expertise. In contrast, their software claims to offer a simpler workflow by breaking down events into “Triggers” and “Effects”, specific measurements or observations that result in specific responses. Moreover, it offers some interesting features like the ability for remote interaction - while I’m not sure how useful this is in practice, the promise of being able to walk away from an experiment and manage it instead through Discord is very cool. The main manuscript mainly focuses on demonstrating two use cases of reactive single cell applications - these both look reasonable to me in my limited expertise as a primarily computational person. I will instead focus my review on the technical components.

Overall, my main concern would be if the image analysis modules in this software are general or not, and the difficulty in creating custom modules. I’m currently under the impression that the moment the user wants to operate on images slightly different from the authors’ set-up, or do tasks other than the arbitrary ones that scripts have been provided for, the software now requires extensive fine-tuning and coding efforts - which seems to contradict the authors’ claims that their software is simpler to work with. It’s possible that I’m missing a key piece of documentation or element of the GUI, and would be open to the authors pointing this out if so.

1. As this software claims to be general, and not an in-house solution, it would be critical to demonstrate that the segmentation and tracking modules are capable of generalizing to images from different groups. The authors fine-tune a pre-trained U-Net model with 50 images from their own lab, and while this might result in a model that works fine in-house for their particular microscopy set-up, I'm not sure if the model is actually general enough to function out-of-the-box on images from different groups without this fine-tuning procedure. I would like to see both quantitative and qualitative results for a variety of other yeast cell images - for example, the benchmark by the Yeast Image Toolkit. It would also be additionally useful if the authors could show generalization to cell markers outside of brightfield images (e.g. cytoplasmic fluorescent images). If it does turn out that annotating images for fine-tuning a model is essential to the performance of the segmentation module, this is a serious limitation that needs to be discussed, since it imposes time and hardware (e.g. a GPU for training) demands, or at least it requires the users to custom-write a module to an alternative segmentation method.

There seems to be a misunderstanding here. We do not claim that we have found a universal solution to solve any image analysis problem in real-time. The message we tried to convey is that if one has or can get an image analysis solution that works on a specific problem in real time, then we propose a software solution that leverages this capacity and the μ manager Python API to orchestrate reactive experiments. We agree that having an image analysis solution that works in real time on the specific problem at hand is not a trivial requirement. However, it is not unrealistic either. Either because the effective task to solve is relatively trivial (detecting a loss of focus, computing the mean pixel fluorescence intensity in a region of interest, etc), or because a deep-learning based solution can be used. In fact, since our target users are primarily quantitative biologists and biophysicists, it is likely that the user has already developed such a solution for the (offline) analysis of their data. Our software significantly decreases the coding effort needed to reuse this tool for online applications. We provided two examples on yeast in a microfluidic device using a DeLTA-based tool. Following the suggestion of the reviewer, we now provide a third one, using Cellpose on bacteria growing on agar pads.

2. Just due to the nature of how diverse these experiments are going to be, it is very likely that users will have to write custom modules. Unfortunately, writing custom modules seems to require highly in-depth knowledge of the code. While the documentation is good at giving a general overview, my main issue is that even accessing and changing basic variables seems to require an exact knowledge of where everything is stored and how they are named. For example, to change the exposure value, I have to know to access the "protocol" object under the micromator core class and know what exact variables to access. Is there any way the authors can streamline or simplify this process (e.g. by having the GUI having a range of standard effects/triggers that the users can parameterize, or by providing better-commented and documented examples that can serve as wrappers - e.g. looping through all cells and calculating X should be a pretty standard thing)? At very least, the in-depth specifics of how parameters are stored and laid out, and what objects control what specifically should be better documented.

It is absolutely true that using MicroMator does necessitate some coding skills and efforts. At the very least, event triggers and effects need to be defined to match the specifications of the reactive experiment of interest. This amounts to editing at least one file, the event creator file, as shown in the example provided in the Supplementary Text (Section 2). This can be relatively trivial if this is close to what has already been done, such as performing image analysis and displaying the number

of cells in the field of view upon image acquisition. Or this can be a relatively substantial task if this involves complex interactions with the microscope, such as moving the stage and depth so as to follow an object. In this case, some knowledge of the μ manager API will be needed too. Advanced use might also require that additional modules are written. This will be the case if some sort of optimization is performed online, other than solving model predictive control problems, or if a novel imaging tool has to be used. Yet, our point is that this will anyway require significantly less coding efforts and expertise than developing from scratch software solutions that robustly solve the same problem. This is an essential point. By providing a rather complete architecture to solve such problems and example solutions to various problems, MicroMator significantly lowers the programming effort needed to design reactive experiment. We clarified the benefit of using MicroMator for the user, and also the need for coding skills at the end of the MicroMator software presentation in the main text.

To get things more concrete for the reader, we provided an example in the Supplementary Text in the initial version of the paper. However, it had no visibility from the main text. Based on the comment of the reviewer, we corrected this mistake and made several references to this example in the main text.

Reviewer #4

While conceptually the reactive control system is brilliant, and the presented data show considerable improvement in achieving target expression (minimizing error), there are some key points that if addressed would make the description of this system clearer.

We thank the reviewer for their positive appreciation of our work.

1- As it can be inferred from fig 2b, the response time (from EL222 being activated to mScarletI being detected) is considerable. This raises the question on how such delay in induction/detection is computed in the feedback information. This is somehow addressed in the supplementary file, where the model is calibrated and validated. Yet, as this is key, it should be better addressed in that section. Particularly, this should be contextualized and compared with other kinetic studies involving EL222 (i.e Benzinger & Mustafa Khammash PMID: 30166548). In addition, according to published data mScarletI has a maturation half-life of 36 min (<https://doi.org/10.1371/journal.pone.0219886>). How this coincide with the kinetics the authors observe and particularly, with the apparent decay of signal ?

This is indeed a point we could have clarified earlier in the main text. As mentioned in the Supplementary Material (Supplementary Note 5), we obtained a good fit to data when using a delay of 36 minutes. We were unaware of the fact that this was precisely the maturation time estimated in the article of McCulloch et al. (2020). We thank the reviewer for this interesting reference. Our estimation is therefore also consistent with the rapid kinetics of the EL222 transcription factor observed by Benzinger and Khammash (2018). We added these facts in the main text and in the supplementary material.

2- While Fig 2B is informative regarding how different duty cycles lead to different degrees of expression, other critical description of the system is missing, regarding the off kinetics: how fast does mScarletI expression levels come down once lights are turned off (and how they go up upon induction, see previous point). All of this is particularly relevant if the

utilized reporter does not have a degradation tag, nor its mRNA has a destabilizing signal. This is not explicitly commented (once again, see i Benzinger & Mustafa Khammash PMID: 30166548).

Our fluorescent protein has no degradation tag, and the mRNA has no destabilization signal. When fitting our model to data, we obtained a good fit with a protein degradation/dilution rate that corresponds to the observed cell growth rate (0.004 min^{-1} , as noted in Tables S1 and S2 and corresponding to a cell generation time of about 180 min). Following the comment of the reviewer, we added this information in the main text.

3- Figure S6A: This inset, containing single cells measurements under defined Light ON:OFF regimes is quite informative on the dynamics of the system and how this can be controlled AND monitored at the single cell level. This dataset could be included as part of Figure 2 (in some way it appears to be more relevant than current 2B)

We agree that the data represented on Fig S6 is providing interesting information on the dynamics of the system. However, the available space for this figure does not allow us to add a panel in it, and we would have to remove the existing panel 2B. And even if, from a system characterization perspective, Fig S6A might indeed be more interesting, the existing Fig 2B exemplifies the fact that several light stimulation profiles can be applied in parallel to cells in the same field of view. We see this possibility as highly attractive from a system identification perspective. Also, we present in Fig S5 all the corresponding data, which is also relatively rich. We modified the caption of this figure (now Figure 3) to highlight the originality of the experiment and we made a reference to Figure S5 for the detailed data.

4- The time scale of the data acquisition (2B, C,D, E) is of 500 mins, which is several times above the typical doubling time of yeast cells (~ 100 mins). Therefore, it is expected that as monitoring each cell, many of them will be budding and diving in several occasions. The latter may create a transient decrease in signal (due to the immediate dilution of mScarletI, and even a prolonged decrease in signal light-activated EL222 is also diluted. Could the authors comment on whether cell division was accompanied with such a decrease in signals, how cell division was accounted for when monitoring single cells, and what happened regarding subsequent decisions (are new daughter cells not considered in further analyses IF expression deviated too much?, etc). While in supplementary material this is indirectly addressed (“When new cells were born, they were assigned a stimulation using our cell-sorter event, such that the population of cells in each bin was approximately even”), the supplementary text would be enriched by addressing these key issues.

Cells are indeed growing and budding during the experiments. The fluorescence that we report corresponds to the mean pixel intensity for each cell. Looking at single-cell data, this signal in mother cells is apparently not affected by budding events. However, we have not performed here a detailed analysis of the impact of budding on intracellular protein concentrations. This would necessitate that mother-daughter relationships are identified with precision, a task that is not always easy in dense monolayers, and would go beyond the scope of our study.

5- Related to the latter point, by looking at video 1 it is clear that, indeed, cells are dividing, yet cells at the edge of the cell cluster are dividing more actively than the ones remaining at the center of the clusters. How does signal intensity vary in the different population/age of cells? Visual inspection of fluorescence (Movie 2) suggests that there may be such a difference, yet looking at Fig 2 or S5 it is not possible to distinguish whether that is the case.

We looked carefully at this (now Movie 3) and other videos, and it was not apparent to our eyes that cell division rates were different inside or at the edge of cell clusters.

To investigate whether the intensities were differing inside or outside the clusters, we manually selected cells located at the center or at the edge of cell clusters, and computed the corresponding fluorescence distributions. The intuition of the reviewer is correct, in the sense that there is indeed a statistical difference between cells at the edge and in the center of the microcolonies when the same light is sent on all cells (Fig 2 left and middle). The reason for this is not obvious for us. However, if the intuition of the reviewer that cells on the edge are growing faster is also correct, then this observation could be explained by a larger protein dilution rate in these cells. Lastly, it is striking that single-cell control was effective in abolishing the existing differences between cells (Fig 2 right).

Fig. 2. Comparison of intracellular fluorescence at the edge of a microcolony and inside a microcolony. (left) open-loop population control experiment (middle) closed-loop population control experiment, and (right) single-cell control experiments. 50 outer and inner cells were measured for each condition.

6- Suppl info: “in which we randomly assigned cells in the field of view to receive 0, 200, 500, or 2000 ms of photostimulation”

Could the authors comment on the intensity of light that was applied, how this compares to other EL222 yeast experiences (i.e Zhao et al PMID: 29562237; Benzinger & Mustafa Khammash PMID: 30166548), and whether intensity can be tuned with MicroMator on individual cells (yet without compromising a given erosion i.e. 33%). Importantly, as in most publications a given light intensity is applied to a cell culture of a particular OD, herein light is directly applied to cells (therefore that should be somehow accounted for)

The two cited papers refer to detailed quantifications of the optogenetic system for cells grown in liquid cultures and stimulated using LEDs. We grow cells in a microfluidic device and stimulate them with an illumination system for fluorescence microscopy. Specifically, we use a pE-4000 CoolLED system and send a 435nm light for short durations with an intensity of 20% of the maximum. Moreover, this light goes through the excitation filter of the CFP cube before reaching the sample (CFP cube: Excitation 436/20 nm, Emission 480/40 nm, Dichroic 455 nm). Even set to a low intensity, it is clear that the light stimulation perceived by cells is much stronger in our setup. Indeed, we achieve near maximal response levels with duty cycles that are less than 1% (1600ms every 3 minutes, see Fig S5), whereas Benzinger et Khammash (2018) managed to get a graded response over the entire range of duty cycles (see eg Fig 2C in their paper). We added a mention to the specific illumination setup we used in the Method section.

Whether DMDs could be used for intensity modulation is a very nice question. Strictly speaking, it is not possible with this technology to send at the same time lights of different intensities in different locations in the field of view. However, the device enables “grayscale” through tilting the mirrors at high frequencies (up to 5000 times per second in principle). Therefore, although this is still duty cycle modulation, this could in practice allow for intensity modulation.

We also note that the duration of the light stimulations we gave in the Supplementary Text were incorrect and have been corrected to match the experimental data presented in Fig S5.

7- Fig S4E: Please make sure the data present in the median Off-target fluorescence is correct.

We checked for this data and it appears correct (the scale of the y-axis is 6 times smaller). However, we realized that the labelling of the y-axes of the plots was not completely clear, and we improved it. This might have led to a confusion.

8- Fig S5: The basal fluoresce (0 ms of photostimulation) is rather high. Could this be due to background environmental light in the place where the microscope is and/or samples are handled? Strictly speaking one would expect the target cells/samples to be in the dark unless they are actively photostimulated. Was this the case, or was background light levels coming from general ceiling lights, monitors, screens etc? (Figure s8)

We agree with the reviewer that the background levels are rather high in the “full dark” experiment, in one replica notably. In particular, we know from bioreactor experiments that background levels can be really low with this system (Aditya et al, 2021). Cells are precultured in the dark, and the microscope is equipped with an opaque environmental box for temperature control. Moreover, the room has black-painted walls and is kept in the dark, with the exception of the low light coming from the computer screens. Yet, to load the cells in the microfluidic device, we use some (low) ambient light. It is therefore likely that cells respond to light stimulations they received during cell loading. This is consistent with the fact that the expression appears to be transient. On this point, we note that bioreactor setups are superior to our CellAsic microfluidic solution since it is relatively easy to do long precultures with bioreactors before the effective start of the experiment.

9- Figure 3. A and B legends are missing in the figure (it appears as the figure is cropped in the upper region)

True. We apologize for the inconvenience. We corrected the issue.

10- Movie S3 and S4: There seems to be two different cell populations, ones with high fluorescence and others with low signals. Could the authors comment on that?

We thank the reviewer for their careful observation of our data and for their comment. Actually, the (rectangular) region that the DMD can illuminate with “all pixels on” is inscribed in the field of view of the microscope and does not fully cover the region recorded by the camera. Consequently, a rectangular region (the topmost 10%) is not illuminated for fluorescence. We have now clarified this in the caption of the movies (Supplementary Movie 3) and annotated the videos.

11- Line 225: “This factor expresses (pATAF1_4x) in turn the mCerulean fluorescent protein fused to a constitutively active Far1 protein (FAR1M_mCerulean)”

- a) Why did the authors express a Far1- mCerulean fusion, and not just mCerulean?
- b) Moreover, why did the authors use a constitutively active Far1?
- c) Whereas methods talks about Far1_mCerulean figure 3 describes it as “mCerulean-Far1M” (c-terminal fusion).
- d) The fact that upon light, CRE is activated, and Far1(+mCerulean) is expressed, can easily explain the slow growth phenotype (see line 144: “Analysis of the lineage trees of targeted cells and non-targeted cells confirmed that recombined cells have a slow growth phenotype”). The authors present this as an almost anecdotal result of recombination, whereas it would be an expected end-result (as Far1 expression should lead to cell cycle arrest). This should be properly explained
- e) What is also confusing is that if the authors decided on purpose to express Far1M, this should have led to a stronger cell cycle arrest, which is not so obvious (as cells continue to divide albeit slower)

Our strain has been designed to exhibit growth arrest upon recombination. With this aim, we expressed a constitutively active form of Far1 upon recombination. This was left implicit in the text and we have now clarified this fact. Moreover, the reviewer rightfully states that the observed slow growth phenotype was indeed fully expected, and that one could even have expected to see a complete cell cycle arrest. Naturally, even if these behaviors were expected, they were not guaranteed to happen, since they depend on quantitative aspects such as the effective expression level of Far1M in our cells, the impact of the fusion with the fluorescent protein, and the time it takes to reach a sufficiently high level of expression. Lastly, as indicated in the Method section, the fusion protein we use is Far1M-mCerulean. The use of “mCerulean-Far1M” in Fig 3 (now Fig 4) is erroneous and has been corrected.

12- Line 238 “Microscopy setup, microfluidics and imaging”

There is no description on the illumination conditions of the MicroMator experimentation room setup, and how “darkness” was achieved when samples (cells) were not being actively illuminated (safety red-lights)? This info is critical as:

- a) the proof of concept figures are based on comparing not illuminated versus illuminated cells
- b) there seems to be a clear background level of mCerulean, which at this point is not clear if it is due to leakiness of EL222 control, or ambient light.
- c) could background EL222 activity (due to ambient light or EL222 leakiness) also lead to cells exhibiting recombination even in the absence of direct illumination?

We could indeed have better documented the experimental setup of our applications. As said above, cells are precultured in the dark, and the microscope is equipped with an opaque environmental box for temperature control. A photography of our experimental setup was shown in Fig S8 (now Fig S9), but it was not referred to from the main text and the caption was minimal. So, we thank the reviewer for this comment and improved our manuscript on these two counts.

Regarding the non-negligible measured levels of blue fluorescence we obtained in the CFP channel for the non-targeted cells in Fig4C, it was not clear whether this corresponded to leaky expression of mCerulean from the pATAF1 promoter or to cell autofluorescence. We therefore compared these fluorescence levels with the ones of wild-type cells. The two distributions appear rather close, with two notable differences. The first one is the fatter right tail of the distribution of the non-targeted cells. As already commented in the main text, this corresponds in part to spontaneous recombination, to inherited recombinations, and to late tracking issues. The second

difference is that the center of the distribution of the non-targeted cells is also slightly right-shifted. This is thus likely that this corresponds to a low leakiness of the pATAF1 promoter. Moreover, the low fluorescence observed in non-targeted cells is often non-homogeneous in the cell (see Supplementary Fig 10). This is consistent with a low expression of Far1M-mCerulean, a fluorescent reporter with nuclear localization.

Fig. 3. Comparison of intracellular fluorescence for wild-type cells (brown) and non-targeted cells (grey). The mean fluorescence for WT cells is 1139 a.u. and the mean fluorescence for non-targeted cells is 1496 a.u.

In our experiments (Fig4C), we do observe recombination in the non-targeted cells. This rate is rather low overall (~25% in the top 10% of the non-targeted cells). Also, it is important to keep in mind that in such experiments, we excite other neighboring cells in the same field of view. So non-targeted cells may receive some residual light (see our quantification of the DMD precision using three color fluorescent microscopy in Supplementary Note 4). Regarding specifically the recombination rate in the dark, it has been quantified by cytometry in Aditya et al (2021; Fig 1C, the genetic construct is similar, although not identical). After 72 hours of growth in the dark, only 0.15% of the cells were found to be recombined. Therefore, recombination due to leakiness of the EL222 promoter is marginal in the dark.

13- The authors do not provide additional information on the reporter mScarletI:

- a) where was it obtained from (addgene, reference etc),
- b) is it yeast optimized ?
- c) does it have a degron sequence?
- d) why was it chosen over other well characterized destabilized reporters?

mScarletI has been codon optimized for expression in *S. cerevisiae* and synthesized by Integrated DNA Technologies (IDT). This is now explicitly mentioned in the text. We chose a red fluorescent protein such that no overlap exists between the excitation spectrum of the fluorophore and the activation spectrum of the optogenetic transcription factor, thus preventing unwanted EL222 activation during fluorescent image acquisition. mScarletI was chosen for its brightness and relatively short maturation time. We have chosen not to use a degradation tag since this would have lowered the fluorescence readout and worsen the signal-to-background ratio.

14- Fig 1. Inset 3. “if n_cells==12 ” is double = correct?

In Python, tests for equality use ==. We replaced it into >=, that might be less confusing.

Reviewers' Comments:

Reviewer #1:

Remarks to the Author:

I had the chance to review a second version of the manuscript "MicroMator: Open and Flexible Software for Reactive Microscopy" by Fox et al. I'm happy to see that the manuscript was improved, e.g. in the context of the description of the software and potential applications. I wasn't sure in the first version who exactly the target audience of the software is and how it looks like in action. The new supplementary material and added text in the manuscript make these aspects very clear. All the supplementary materials appear a treasure for potential new users to explore what MicroMator is capable of doing.

I would like to add two minor suggestions.

* In the introductory section, in lines 41/42 "...image analysis has recently made a giant leap in terms of accuracy and rapidity thanks to deep learning methods..." I do agree with this statement and think putting a citation here makes sense. The authors may want to take a look at <https://www.nature.com/articles/s41592-018-0267-9> to get an overview about works which argue in a similar direction.

* Coming back to my suggestion in the first round, I would remove the word "simple" in the description of the graphical user interface in line 81. In supplementary movie 1 at minute 3:00, it appears obvious to me that the user-interface is not simple. It is a complex interface spanning half of the screen with a lot of text fields and buttons which are not self-explanatory to a new user.

Sincerely,
Robert Haase

robert.haase@tu-dresden.de

Reviewer #2:

Remarks to the Author:

The authors have very nicely addressed all my comments and I recommend publication.

Reviewer #3:

Remarks to the Author:

The authors have fully addressed my comments - my main issue is that I was unclear that MicroMator was intended to be a programming framework, as opposed to an end-to-end general solution, a confusion that I see I shared with Reviewers 1 and 2. The rewritten manuscript makes this much clearer and includes information about the target audience and caveats. That being said, now that the authors have made this clearer, I am skeptical that MicroMator will provide a significant time or expertise savings for users. For example, several reviewers including myself expressed concerns about the generality of the built-in segmentation tools. It's now become clear that if a user needs to use a different segmentation module, which is very likely given most segmentation methods lack generality across experiments/modalities, they will have to code it for themselves and figure out how to get it to work as a module in MicroMator. I'm not fully convinced of why I should use MicroMator over just coding an in-house solution if I already have the programming expertise. That being said, I recognize that others might have a preference for a scaffold/framework, so I think having more tools around is not a bad thing, so I recommend acceptance at this stage of the manuscript.

Reviewer #4:

Remarks to the Author:

This reviewer thanks the authors for the detailed rebuttal to the previously addressed points regarding the EL222 system and aspects of the optogenetic setup and related topics.

The previous doubts/reservations have been satisfactorily addressed. This appears to be nice work that will help to advance optogenetic manipulation with great precision and accuracy! Nice

Reviewer #1

I had the chance to review a second version of the manuscript "MicroMator: Open and Flexible Software for Reactive Microscopy" by Fox et al. I'm happy to see that the manuscript was improved, e.g. in the context of the description of the software and potential applications. I wasn't sure in the first version who exactly the target audience of the software is and how it looks like in action. The new supplementary material and added text in the manuscript make these aspects very clear. All the supplementary materials appear a treasure for potential new users to explore what MicroMator is capable of doing.

We thank the reviewer for their positive appreciation of our work.

I would like to add two minor suggestions.

* In the introductory section, in lines 41/42 "...image analysis has recently made a giant leap in terms of accuracy and rapidity thanks to deep learning methods..." I do agree with this statement and think putting a citation here makes sense. The authors may want to take a look at <https://www.nature.com/articles/s41592-018-0267-9> to get an overview about works which argue in a similar direction.

We thank this reviewer again for their suggestion, and we have added a citation to the proposed article to provide support for our assertion in lines 41/42.

* Coming back to my suggestion in the first round, I would remove the word "simple" in the description of the graphical user interface in line 81. In supplementary movie 1 at minute 3:00, it appears obvious to me that the user-interface is not simple. It is a complex interface spanning half of the screen with a lot of text fields and buttons which are not self-explanatory to a new user.

We understand the reviewer's point of view that simple is indeed a subjective description pertaining to user interfaces and have removed the word "simple" from line 81.

Sincerely,
Robert Haase
robert.haase@tu-dresden.de

Reviewer #2

The authors have very nicely addressed all my comments and I recommend publication.

We thank the reviewer for their positive appreciation of our work.

Reviewer #3

The authors have fully addressed my comments - my main issue is that I was unclear that MicroMator was intended to be a programming framework, as opposed to an end-to-end general solution, a confusion that I see I shared with Reviewers 1 and 2. The rewritten

manuscript makes this much clearer and includes information about the target audience and caveats. That being said, now that the authors have made this clearer, I am skeptical that MicroMator will provide a significant time or expertise savings for users. For example, several reviewers including myself expressed concerns about the generality of the built-in segmentation tools. It's now become clear that if a user needs to use a different segmentation module, which is very likely given most segmentation methods lack generality across experiments/modalities, they will have to code it for themselves and figure out how to get it to work as a module in MicroMator. I'm not fully convinced of why I should use MicroMator over just coding an in-house solution if I already have the programming expertise. That being said, I recognize that others might have a preference for a scaffold/framework, so I think having more tools around is not a bad thing, so I recommend acceptance at this stage of the manuscript.

We thank the Reviewer for their comments and appreciate different perspectives on this point. We are glad that the revised version has at least made it clear what the intention of MicroMator is.

Reviewer #4

This reviewer thanks the authors for the detailed rebuttal to the previously addressed points regarding the EL222 system and aspects of the optogenetic setup and related topics. The previous doubts/reservations have been satisfactorily addressed. This appears to be nice work that will help to advance optogenetic manipulation with great precision and accuracy! Nice

We thank the reviewer for their positive appreciation of our work.